



# 1 A numerical analysis of biogeochemical controls with physical

# 2 modulation on hypoxia during summer in the Pearl River Estuary

Bin Wang [1] , Jiatang Hu [1,*] , Shiyu Li [1,*] , Dehong Liu [1]
[1] Guangdong Provincial Key Laboratory of Environmental Pollution Control and Remediation Technology (Sun
Yat-sen University), School of Environmental Science and Engineering, Sun Yat-sen University, Guangzhou, 510275,
China
Correspondence to: Jiatang Hu (jiatanghu@126.com), Shiyu Li (eeslsy@mail.sysu.edu.cn)
**Abstract.**
As an important biogeochemical indicator of aquatic ecosystem, dissolved oxygen (DO) is affected by the
boundary conditions and biogeochemical processes. Biogeochemical processes can affect DO concentrations by
directly consuming or generating oxygen locally, or through changing the DO fluxes from the ambient water bodies.
However, the latter mechanism is still unclear. In this study, a novel method named physical modulation of
biogeochemical terms is therefore proposed and coupled to a physical-biogeochemical model to investigate their
contributions to the hypoxia during the summer of the Pearl River Estuary (PRE).
According to the result of modulation method, re-aeration and sediment oxygen demand are the most important
biogeochemical processes, and determine the distribution, the spatial extent, and the duration of hypoxia in the PRE. A
DO balance analysis is conducted and reveals that although the re-aeration occurs on the air-sea interface, the
reoxygenation leads to a strong DO gradient form between the surface and lower layers. As a result, the majority (89 %)
of oxygen entering the surface layer from the atmosphere will be transported to the lower layer through the vertical
diffusion, and 28 % eventually reach the bottom layer. Similarly, after consuming the bottom DO, sediment oxygen
demand facilitates the downward DO flux of vertical diffusion and decreases the upward DO flux of vertical advection.
Under the modulation of physical processes, sediment oxygen demand causes a most significant decrease in DO
concentration by 4.31 mg L$^{-1}$ in the bottom of the HFZ (a high frequency zone of hypoxia located off the Modaomen
sub-estuary) and the west of lower estuary. However, the re-aeration supplements an average of 4.84 mg L$^{-1}$ DO on the
west of lower estuary, which leads to hypoxia only occur in HFZ. Numerical experiments show that turning off the
re-aeration leads to an expansion of hypoxic area from 237 km$^2$ to 2203 km$^2$ and results in a shift of hypoxic center to
the west of lower estuary. Moreover, a persistent hypoxia (hypoxic frequency>80 %) is observed in the west of lower
estuary. When compared with re-aeration and sediment oxygen demand, photosynthesis and water column respiration
have fewer effects on DO conditions. In the bottom of the HFZ, photosynthesis exceeds the water column respiration
and eventually supplements DO concentration by 0.98 mg L$^{-1}$, causing an increase of hypoxic area to 591 km$^2$.



## 1. Introduction

Since dissolved oxygen (DO) is essential for the survival of aerobic aquatic organisms, hypoxia, a condition where the water body is deprived of adequate oxygen is detrimental to the aquatic ecosystems in terms of the behavioral, physiological, and productive impacts, such as reduced growth, mortality, and loss of reproductive capacity (Rabalais et al., 2010). In recent decades, hypoxia has been dramatically exacerbated by human activities. As a result, there have been over 400 coastal hypoxic zones covering more than 245,000 km$^2$ in the world (Diaz and Rosenberg, 2008).

The formation and maintenance of hypoxia is related to the interactions of physical and biogeochemical processes. Therefore, human activities that change biogeochemical processes affect DO conditions. For example, the excessive nutrient loads will stimulate the growth of phytoplankton and hence promote photosynthesis to produce more oxygen. However, phytoplankton will also use more oxygen for respiration to meet its growth. Moreover, the debris generated after the death of phytoplankton will deposit to the sediment along with terrestrial particulate organic matter and form the sediment oxygen demand. The statistical linkage between nutrient loads and the spatial extent of hypoxia is well documented in the Chesapeake Bay (Hagy et al., 2004) and the northern Gulf of Mexico (Justic et al., 2003). In the Pearl River estuary, a study conducted by Zhang and Li (2010) illustrates the input of terrestrial particulate organic matter will affect sediment oxygen demand, and hence affect DO concentration. Physical processes affect DO conditions by changing the horizontal and vertical DO transport. For example, the hypoxia in the Changjiang estuary is related to the inflow of Taiwan Warm Current with low-oxygen level (Wang, 2009;Wang et al., 2012). In the Chesapeake Bay, the summer hypoxia is caused by the stratification which inhibits the downward DO transport through the vertical diffusion (Du and Shen, 2015). In addition, physical processes can regulate DO concentration by changing the nutrient distributions. In some coastal areas, the upwelling will bring the bottom nutrients to the surface layer and promote the primary productivity, followed by the occurrence of hypoxia (Rabalais et al., 2010).

Since physical and biogeochemical processes are highly interacted, it is essential to distinguish their contributions to the DO conditions. In the Chesapeake Bay, Shen et al. (2013) use two timescales to quantify the physical and biogeochemical processes concerning the DO conditions. Accordingly, they suggest that the physical processes accounts for 88 % of variations in DO distribution in the hypoxic zone (Du and Shen, 2015). In addition, the DO budget analysis is commonly used to investigate the contributions of physical and biogeochemical processes to changes of DO (Scully, 2010;Montes et al., 2014;Li et al., 2015;Yu et al., 2015). Biogeochemical processes exert effects on DO conditions by two mechanisms, one is by directly consuming or producing oxygen locally, and the other is by changing the DO fluxes from the ambient water bodies. The latter mechanism is understood as the contributions of ambient biogeochemical processes on DO concentrations. Take the re-aeration as an example, in spite of its occurrence on the air-sea interface, the oxygen entering the surface layer through the re-aeration will be transported to lower layers and change DO concentration. Given this mechanism remains unclear, it is necessary to be investigated since it may play an important role in DO conditions and provide a further insight into the DO dynamics.

The Pearl River is the second largest river in terms of the river discharge with an annual averaged discharge of 10,524 m$^3$ s$^{-1}$, among which 80 % is delivered during the wet season (Ou et al., 2009;Zhang and Li, 2010). The river



network includes Beijiang (North River), Xijiang (West River), Dongjiang (East River), Liuxi River, and Tan River,
covering a drainage of $4.5*10^5 \, km^2$. Fresh water from the river network is emptied into the Northern South China Sea
(NSCS) through the eight outlets, namely Humen, Jiaomen, Hongqili, Hengmen, Modaomen, Jitimen, Hutiaomen, and
Yamen. The Pearl River Estuary (PRE) is located on the Northern South China Sea and adjacent to the Pearl River
network. The PRE consists of four sub-estuaries, including the Lingdingyang, Modaomen, Jitimen, and Huangmaohai
sub-estuary, among which the Lingdingyang is the most principal and largest estuary with an area of almost 2,000 $km^2$.
The PRE is a complex estuarine system characterized by the shallow bank in the west of the estuary with the depth of
less than 5 m and two deep channels with the depth more than 10 m and the width of 1km. In the summer, the physical
processes is influenced by the huge river discharge and southeasterly wind. Thus, the nutrient and DO distributions are
influenced by the combination of complicated dynamical and topographic characteristics to a large extend. Since the
1970s, several large-scale field observations have reported the seasonal hypoxia in the PRE (Yin et al., 2004;Ye et al.,
2013). Some studies have also been conducted to use water quality models to reproduce and investigate the hypoxia in
the PRE (Guan et al., 2001a, b;Luo et al., 2008). Previous studies suggest that sediment oxygen demand and
stratification are two main reasons for the hypoxia in the PRE (Yin et al., 2004;Zhang and Li, 2010). However, unlike
the northern Gulf of Mexico, hypoxia in the PRE is intermittent and confined to a small scale (Rabouille et al., 2008),
though the nutrient loadings in the PRE are similar to those in the Mississippi River (Hu and Li, 2009). In general,
although the hypoxia in the PRE has been observed for many years, the mechanisms remains unclear.
Motivated by these previous studies, the purpose of this study is introducing the physical modulation of
biogeochemical terms to investigate the characteristics of hypoxia in the PRE, including the distribution of hypoxia,
the major processes controlling DO balance, and the reasons for why the hypoxia in the PRE occurs in the specific area
and is not severe. The manuscript is organized as follows. In section 2, we describe the physical and water quality
model used in this study, as well as the theory and methodology of modulation method. In section 3, we validate the
coupled model and the modulation method. Section 4 provides the results and discussions. Summaries and conclusions
are given in section 5.
**2. Methods**
2.1 Physical model
In order to accurately simulate the dynamic processes forced by a multichannel river network, we use a 1-D and
3-D coupled physical model (Hu and Li, 2009;Hu et al., 2011) which integrates the Pearl River network, the Pearl
River Estuary (PRE), and its adjacent coastal waters in an overall modeling system. Specifically, a 1-D river network
model is dynamically coupled with a 3-D coastal model for the PRE using an explicit coupling approach. The eights
river outlets (see Fig. 1a) serve as the coupling interface between the 1-D and the 3-D model domains. These two
models run in parallel, and their model quantities are exchanged across the coupling interface during runtime. At each
time-step, the 3-D model utilizes the simulated discharge obtained from the 1-D model as the river boundary forcing
and the 3-D model sends simulated water levels to the 1-D model as the downstream boundary forcing for the next
time-step. A detailed description on the methodology and implementation of the coupled model can be found in Hu and



Li (2009).
The 1-D model uses a Preissmann implicit scheme and an iterative approach to solve the Saint Venant equations
of mass and momentum conservation. A salinity transport module is also incorporated in the model. For details on the
1-D model and its governing equations, see Hu and Li (2009). The 1-D model simulates 299 major branches of the
river network with 1726 cross-sections and 189 nodes (see Fig. 1b). For the upstream boundaries, the real-time river
discharge or water levels with zero salinity are specified at Shizui, Gaoyao, Shijiao, Laoyagang, and Boluo (see Fig. 1a
for their locations). The initial conditions of water level and salinity are set to be zero homogeneously. The time step is
5 s for the 1-D hydrodynamic.
The 3-D model utilized is the Estuaries and Coastal Ocean Model with Sediment Module (ECOMSED)
(HydroQual, 2002), which has been extensively used in estuaries. The model solves the Navier-Stokes equations with
hydrostatic and Boussinesq approximations. A Smagorinsky-type formula is applied to parameterize horizontal mixing,
and the Mellor-Yamada turbulent closure model is applied to calculate vertical viscosity and diffusivity (Mellor and
Yamada, 1982). Details on the ECOMSED and its governing equations can be found in HydroQual (2002). The 3-D
model has 183×186 horizontal grids with resolutions that vary from approximately 400-500 m inside the estuary to 3-4
km near the open boundary (see Fig. 1b). Vertically, there are 16 sigma levels with refined resolution in the surface and
bottom. At the open boundaries, tide forcing is prescribed using water levels derived from the Oregon State University
Tidal Data Inversion Software (OTIS), with the uniform salinity and temperature boundary conditions using the
observed data from Wanshan Island (Hu and Li, 2009). Atmosphere forcing is interpolated into our model grid using
wind, air pressure, and net solar radiation obtained from the ERA-interim (the Interim ECMWF Re-Analysis,
http://www.ecmwf.int/en/research/climate-reanalysis/era-interim). The spatial resolution of ERA-interim data is
0.125º*0.125º and the temporal resolution is 6 hours. Initial conditions of water level and salinity are set to be zero
homogeneously. The time step is 5 s and 30 s for external mode and internal mode, respectively.
The 1-D and 3-D coupled physical model is first implemented for November and December 2005 to reach a
steady state. It is then continuously run from January 1st 2006 to December 31st 2006. The results of July and August
2006 are obtained for further analysis.
2.2  Water quality model
In this study, the water quality model used is the Row-Column Aesop (RCA). RCA is developed by HydroQual
(HydroQual, 2004) and is able to directly interface with ECOM and ECOMSED. In the water column, RCA can
simulate five interacting systems including the carbon cycle (C), the nitrogen cycle (N), the phosphorus cycle (P), the
silicon cycle (Si), and dissolved oxygen (DO) (see Fig. 2a). In addition, a sediment flux module is incorporated to
RCA, which simulates the depositional flux of particulate organic matter (POM, including PON, POP, and POC),
diagenesis process in the sediment converting POM to dissolved matters, and transportation of dissolved matter from
sediment to overlying water (see Fig. 2b). Interactions between water column and sediment can be simulated internally
by the RCA. At each time-step, the water quality module calculates the depositional flux of POM for the sediment flux
model. And at the same time, the sediment flux model sends simulated sediment oxygen demand and fluxes in
nutrients for the water quality module as the bottom boundary conditions.





In the wet seasons of the PRE, the high concentration of sediment will limit the growth of phytoplankton by
reducing the light penetration in the water column. In this case, the RCA is modified to simulate the shading effects of
sediment (Toro, 1978):
$Ke = 0.052N + 0.174D + 0.031P$ (1)
Where $Ke$ represents the light extinction coefficient (1 m$^{-1}$), $N$ represents the sediment concentration (mg L$^{-1}$), $D$
represents the concentration of POM (mg L$^{-1}$), and $P$ represents the concentration of Chlorophyll-α (μg L$^{-1}$). The
concentration of sediment is simulated by a sediment transport module which is incorporated to our physical model
(HydroQual, 2002;Hu and Li, 2009).
The equation for each water quality variable is given by:
$$\frac{\partial C}{\partial t} = \frac{\partial}{\partial x}\left(E_x \frac{\partial C}{\partial x}\right) + \frac{\partial}{\partial y}\left(E_y \frac{\partial C}{\partial y}\right) + \frac{\partial}{\partial z}\left(E_z \frac{\partial C}{\partial z}\right) - u\frac{\partial C}{\partial x} - v\frac{\partial C}{\partial y} - w\frac{\partial C}{\partial z} \pm S - W$$ (2)
Where $C$ represents concentrations of each water quality variables. $x, y,$ and $z$ represent the two horizontal
coordinates and single vertical coordinate. $u, v,$ and $w$ represent velocity components in the $x, y,$ and $z$
coordinates, respectively. $E_x, E_y,$ and $E_z$ represent dispersion coefficients. The $S$ parameter represents sources and
sinks. $W$ represents external inputs of nutrients and oxygen-demanding materials which come from municipal and
industrial discharges, river discharges, and atmospheric deposition.
For the dissolved oxygen, the sources are re-aeration (Rea) for the air-sea interface and photosynthesis (Phot), the
sinks are nitrification (Nitri) of ammonia, oxidation (Oxid) of dissolved organic matter and dissolved sulfide,
respiration (Resp) by phytoplankton, and sediment oxygen demand (SOD) for the water-sediment interface. In this
study, we combine nitrification, oxidation, and respiration into water column respiration (WCR) to represent the total
DO depletion in the water column. The equation describing these kinetic processes is given as:
$S_{DO} = k_a \theta_a^{T-20}(DO_{sat} - DO) + \alpha_{OC} \cdot \alpha_{NH_4} \cdot G_P \cdot P_c + (\alpha_{NO_{23}c}) \cdot (1 - \alpha_{NH_4}) \cdot G_P \cdot P_c$
$\quad - 2 \cdot \alpha_{ON} \cdot k_{14,15}\theta_{14,15}^{T-20} \cdot NH_4 \cdot \dfrac{DO}{K_{nitri} + DO}$
$\quad - \alpha_{OC} \cdot \left[ k_{20,0}\theta_{20,0}^{T-20} \cdot RDOC + k_{21,0}\theta_{21,0}^{T-20} \cdot LDOC \cdot \dfrac{LDOC}{K_{LDOC} + LDOC} + k_{22,0}\theta_{22,0}^{T-20} \cdot ReDOC \cdot \dfrac{ReDOC}{K_{LDOC} + ReDOC} \right.$
$\quad \left. + k_{23,0}\theta_{23,0}^{T-20} \cdot ExDOC \cdot \dfrac{ExDOC}{K_{LDOC} + ExDOC} \right] \cdot \dfrac{P_c}{K_{Pc} + P_c} \cdot \dfrac{DO}{K_{DO} + DO}$
$\quad - \alpha_{OC} \cdot k_{PR}(T) \cdot P_c$
$\quad - k(DO_{sed} - DO)$
$\quad - k_{O_2^*}\theta_{O_2^*}^{T-20} \cdot O_2^* \cdot \dfrac{P_c}{K_{Pc} + P_c} \cdot \dfrac{DO}{K_{DO_{O_2^*}} + DO}$ (3)
Where $k_a$ represents the surface mass transfer coefficient (m day$^{-1}$) for re-aeration; $\theta_a$ , $\theta_{14,15}, \theta_{20,0}, \theta_{21,0}, \theta_{22,0},$
$\theta_{23,0}, \theta_{O_2^*}$ represent temperature coefficient; $DO_{sat}$ represents saturation concentration of dissolved oxygen (mg O
L$^{-1}$); $\alpha_{OC}$ represents oxygen to carbon ratio; $\alpha_{NH_4}$ represents preference for ammonium uptake by phytoplankton;
$G_P$ represents specific phytoplankton growth rate (day$^{-1}$); $P_c$ represents phytoplankton biomass (mg C L$^{-1}$); $\alpha_{NO_{23}c}$



represents oxygen to carbon ratio for nitrate uptake; $\alpha_{ON}$ represents oxygen to nitrogen ratio; $k_{14,15}$ represents
nitrification rate at $20^O$C (day$^{-1}$); $K_{nitri}$, $K_{DO}$, $K_{DO_{O_2^*}}$ represents half saturation constant for oxygen limitation (mg O
L$^{-1}$); $k_{20,0}$, $k_{21,0}$, $k_{22,0}$, $k_{23,0}$ represents oxidation rate for RDOC, LDOC, ReDOC, and ExDOC at $20^O$C (day$^{-1}$),
whereby RDOC, LDOC, ReDOC, and ExDOC represent labile dissolved organic carbon, refractory dissolved organic
carbon, reactive dissolved organic carbon, and algal exudate dissolved organic carbon; $K_{LDOC}$ represents Michaelis
constant for LDOC (mg C L$^{-1}$); $K_{Pc}$ represents half saturation constant for phytoplankton limitation (mg C
L$^{-1}$); $k_{PR}(T)$ represents temperature corrected algal respiration rate (day$^{-1}$); $k$ represents transfer coefficient between
the sediment and overlying water; $DO_{sed}$ represents concentration of dissolved oxygen in the sediment (mg O L$^{-1}$);
and $k_{O_2^*}$ represents oxidation rate of dissolved sulfide.

10       The simulation periods of water quality model are the same as the physical model with a time-step of 30 s. Initial

conditions are derived from a 61-days spin up simulation. The initial conditions are replaced by the results at Day 61
and then we run the water quality model again. These processes repeat for 3 times to reach a steady state. The river
boundaries of water quality variables are based on monthly observed data from 2006 collected by the State Oceanic
Administration (including DO, NH$_4$, NO$_2$+NO$_3$, and PO$_4$) as well as the studies conducted by Liu et al. (2015). The
open boundaries of water quality variables are specified as a constant according to observed data obtained from the
State Oceanic Administration and a study conducted by Zhang and Li (2010). The pollutant data from streams, waste
water treating plant (WWTP), and factories which discharge waste water into the estuary directly are provided by the
Shenzhen Environmental Protection Monitoring Center, the Environment Council of Macau Special Administrative
Region (REAM), and the Environment Protection Department of the Hong Kong Special Administrative Region. The
primary parameters in the water quality model are based on previous studies in the PRE (see table. 1)
2.3 Physical modulation of biogeochemical terms

22       Since DO concentration is affected by boundary conditions and biogeochemical processes, the DO flux

transported by dynamical processes actually contains two kinds of effects originating from these two processes.
However, since traditional DO balance analysis does not distinguish between these two factors, we propose a method
named the physical modulatation of biogeochemical terms to simulate these two processes and investigate the
contributions of these two processes to DO conditions. The method assumes that DO can be divided into two separated
parts, including the simulated DO concentration forced by either boundary conditions ($DO_{BC}$) or biogeochemical
processes ($DO_{Bio}$, the increase or decrease in DO concentration due to the effects of biogeochemical processes).
$DO = DO_{BC} + DO_{Bio}$    (4)
Equations of DO, $DO_{BC}$, and $DO_{Bio}$ can be given as:
$\dfrac{\partial DO}{\partial t} + ADV(DO) - DIFF(DO) = \pm S$    (5)
$\dfrac{\partial DO_{BC}}{\partial t} + ADV(DO_{BC}) - DIFF(DO_{BC}) = 0$    (6)




$$\frac{\partial DO_{\mathrm{Bio}}}{\partial t} + ADV(DO_{\mathrm{Bio}}) - DIFF(DO_{\mathrm{Bio}}) = \pm S \qquad (7)$$
Where ADV represents the process of advection ($u\frac{\partial}{\partial x} + v\frac{\partial}{\partial y} + w\frac{\partial}{\partial z}$); DIFF represents the process of diffusion
($\frac{\partial}{\partial x}\left(E_x\frac{\partial}{\partial x}\right) + \frac{\partial}{\partial y}\left(E_y\frac{\partial}{\partial y}\right) + \frac{\partial}{\partial z}\left(E_z\frac{\partial}{\partial z}\right)$); and $S$ represents biogeochemical sources and sinks which are calculated with
reference to Eq. (3), including re-aeration, photosynthesis, water column respiration, and sediment oxygen demand.
Therefore the $DO_{Bio}$ can be estimated by:
$$DO_{\mathrm{Bio}} = DO_{\mathrm{Rea}} + DO_{\mathrm{Phot}} + DO_{\mathrm{WCR}} + DO_{\mathrm{SOD}} \qquad (8)$$
Where $DO_{\mathrm{Rea}}$, $DO_{\mathrm{Phot}}$, $DO_{\mathrm{WCR}}$, and $DO_{\mathrm{SOD}}$ represent the increase or decrease in DO concentration due to the
effects of re-aeration, photosynthesis, water column respiration, and sediment oxygen demand, respectively. These four
variables are simulated according to Eq. (7) except that $\pm S$ represents each corresponding biogeochemical terms,
respectively. The negative values of $DO_{\mathrm{WCR}}$, and $DO_{\mathrm{SOD}}$ indicate that water column respiration and sediment oxygen
demand are oxygen-consuming processes. The detailed derivations of Eq. (4) are given as follows.
According to the mathematical induction, we assume the Eq. (4) is satisfied in the time step i:
$$DO_{\mathrm{i}} = DO_{\mathrm{BC_i}} + DO_{\mathrm{Bio_i}} \qquad (9)$$
Then the DO concentration, $DO_{\mathrm{BC}}$, and $DO_{\mathrm{Bio}}$ in the time step i+1 can be calculated by discretizing the Eq. (5)-(7):
$$DO_{\mathrm{i+1}} = DO_{\mathrm{i}} - \Delta t \times ADV(DO_{\mathrm{i}}) + \Delta t \times DIFF(DO_{\mathrm{i}}) \pm \Delta t \times S \qquad (10)$$
$$DO_{\mathrm{BC_{i+1}}} = DO_{\mathrm{BC_i}} - \Delta t \times ADV(DO_{\mathrm{BC_i}}) + \Delta t \times DIFF(DO_{\mathrm{BC_i}}) \qquad (11)$$
$$DO_{\mathrm{Bio_{i+1}}} = DO_{\mathrm{Bio_i}} - \Delta t \times ADV(DO_{\mathrm{Bio_i}}) + \Delta t \times DIFF(DO_{\mathrm{Bio_i}}) \pm \Delta t \times S \qquad (12)$$
Substituting Eq. (9) into Eq. (10), the DO concentration can be represented as:
$$DO_{\mathrm{i+1}} = \left(DO_{\mathrm{BC_i}} + DO_{\mathrm{Bio_i}}\right) - \Delta t \times ADV\left(DO_{\mathrm{BC_i}} + DO_{\mathrm{Bio_i}}\right)$$
$$+ \Delta t \times DIFF\left(DO_{\mathrm{BC_i}} + DO_{\mathrm{Bio_i}}\right) \pm \Delta t \times S$$
$$= DO_{\mathrm{BC_i}} - \Delta t \times ADV\left(DO_{\mathrm{BC_i}}\right) + \Delta t \times DIFF\left(DO_{\mathrm{BC_i}}\right)$$
$$+ DO_{\mathrm{Bio_i}} - \Delta t \times ADV\left(DO_{\mathrm{Bio_i}}\right) + \Delta t \times DIFF\left(DO_{\mathrm{Bio_i}}\right) \pm \Delta t \times S$$
$$= DO_{\mathrm{BC_{i+1}}} + DO_{\mathrm{Bio_{i+1}}} \qquad (13)$$
Thus it can be concluded that in each time step, DO concentration satisfies the Eq. (4). Furthermore, the DO
increments can be divided into three parts as Eq. (14) shows:
$$\Delta DO = \Delta DO_{\mathrm{BC}} + \Delta DO_{\mathrm{Bio}}$$
$$= \Delta t \times \left[-ADV(DO_{\mathrm{BC}}) + DIFF(DO_{\mathrm{BC}})\right]$$
$$+ \Delta t \times \left[-ADV(DO_{\mathrm{Bio}}) + DIFF(DO_{\mathrm{Bio}})\right]$$
$$\pm \Delta t \times S \qquad (14)$$
Where $\Delta t \times \left[-ADV(DO_{\mathrm{Bio}}) + DIFF(DO_{\mathrm{Bio}})\right]$ represents the contributions of ambient biogeochemical processes.
This term indicates the mechanism that biogeochemical processes can indirectly affect the DO concentration by
changing DO fluxes. $\Delta t \times \left[-ADV(DO_{\mathrm{BC}}) + DIFF(DO_{\mathrm{BC}})\right]$ and $\pm \Delta t \times S$ represent the contributions of boundary
conditions, and local biogeochemical processes, respectively. It is worth mentioning that, in previous studies, the sum
of $\Delta t \times \left[-ADV(DO_{\mathrm{BC}}) + DIFF(DO_{\mathrm{BC}})\right]$ and $\Delta t \times \left[-ADV(DO_{\mathrm{Bio}}) + DIFF(DO_{\mathrm{Bio}})\right]$ is regarded as physical





processes for DO transport.
In this paper, we add five additional variables to the water quality model (RCA), namely $DO_{BC}$, $DO_{Rea}$, $DO_{Phot}$,
$DO_{WCR}$, and $DO_{SOD}$. The same initial and boundary conditions are used for computing $DO_{BC}$ as used for DO
simulations. $DO_{Rea}$, $DO_{Phot}$, $DO_{WCR}$, and $DO_{SOD}$ are set to be zero for initial and boundary conditions. The $\pm S$
represents each biogeochemical process associated with DO and is calculated at each time step by Eq. (3). In addition,
further validations of this modulation method against model results will be given out in the following section.
**3.   Model validation**
3.1  validations of physical and water quality models
Data sets used for model validation include hourly water level data from 8 tidal gauge stations and cruise
observations conducted in July and August 2006 (see Fig. 3a). These tidal gauge stations are located in Jiaomen,
Hengmen, Modaomen, Jitimen, Hutiaomen, Yamen, Zhuhai, and Wanshan Island. The cruise data set includes profiles
of salinity (black circles), temperature (black circles), and dissolved oxygen (DO) (red crosses).
As shown in Fig. 3b, the Taylor diagram shows a statistical evaluation of our physical and water quality coupled
model in terms of dynamical variables (e.g. water level, salinity, and temperature) and DO. Grey isolines provide a
measure of skill, which is represented by centered root-mean-square difference (RMSD) normalized by the observed
values. The distance between the observed point (red pentagram) and each simulated point is proportional to the
RMSD. The angular coordinate gives the magnitude of correlation with observations, and the radial coordinate
represents standard deviation of both observed and simulated values. The observation represents the perfect model
skills to reproduce observations with correlation 1, normalized RMSD 0, and normalized standard deviation 1.
The validation indicates that our coupled model is robust to simulate both dynamical and biogeochemical
processes regardless of their complexity. Specifically, the model simulates water levels at eight tidal gauge stations
(red triangles) and salinity (orange diamonds) distribution well, since the normalized RMSD is considerably small
(<0.40 of standard deviation of observation) and the correlation is high (>0.90). Furthermore, the normalized standard
deviations of both water levels and salinity are clear to 1, which indicates the model reproduces a similar range of
water levels and salinity with observations. The model underestimates the range of temperature (blue circle) with the
normalized standard deviation 0.75. The same is true for DO simulation (green square) with the normalized standard
deviation almost 0.63 since the model does not capture the observed super saturation of DO in the surface and
generally overestimates DO concentration in the bottom. However, the correlation of DO is still relatively high and the
normalized RMSD is within 0.80 of standard deviation of observations. To further gain an insight into the difference
between the simulation and observations, we analyze the frequency distribution of the biases which is normalized by
the standard deviation of observations (see Fig. 3c, d). Generally, 85 % of normalized biases are within $\pm 1$ and the
coupled model underestimates DO concentration by 0.34 of standard deviation of observation.
Since we are most interested in the DO concentration, the comparison between the simulated and observed DO
concentration in the bottom is shown in Fig. 4. Each cruise samples stations over a 3 or 4 day period. The simulated
DO concentration used for the comparison are averaged over the same period. Despite the complexity, our model





reproduces the observed spatial distribution in DO concentrations and captures the observed hypoxia (see Fig. 4c) on
the shelf off the Modaomen sub-estuary. The DO concentration is high in the upper reaches of the estuary and
increases gradually along the estuary to a value of 5mg L$^{-1}$ in the lower estuary. This low DO concentration in the
upper reaches of the estuary is due to the low DO concentration discharged from the river outlets.

5       With quality control, a comparison between the simulated and historical estimated summer re-aeration, sediment

oxygen demand, and respiration by phytoplankton in the Lingdingyang Bay is shown in Table 2. The simulated values
are in reasonable agreement with the estimations and furthermore are comparable to the historical estimated
distributions. The re-aeration DO replenishment rates show strong spatial variability, with the maximum values near
the river outlets, and decreases sharply to negative in the mouth of the estuary. The values of the sediment oxygen
demand reach their maximum values in the middle of the estuary.
3.2  validations of physical modulation

12       In order to evaluate the accuracy of physical modulation to simulate the DO concentration, comparisons of the

two-month averaged DO concentration simulated by the water quality model and modulation method in the surface,
middle, and bottom layer are shown in Fig. 5. Overall, the DO distributions simulated by the modulation method are in
good agreement with those simulated by the water quality model. In the surface, the modulation method generally
overestimates RCA simulations in the estuary and its adjacent areas while underestimates RCA simulations on the shelf
(see Fig. 5a). This is the same true for the middle layer (see Fig. 6b). Figure 6c reveals that in the bottom layer, the
modulation method simulates slightly higher DO concentrations within the whole model domain. To gain further
insight in to the differences in DO concentration of the modulation method and water quality model, we analyze the
volume distribution of the biases (see Fig. 6a). The distributions as a metric for quantifying biases are computed over
the whole model domain and for different bias bins between -1.0 and 1.0 mg L$^{-1}$. According to Fig. 6a, the layers
where bias varies between -0.1and 0.3 mg L$^{-1}$ occupy about 97 %. Thus, the DO concentration simulated by
modulation method is close to the one derived from RCA simulations whereas it is overestimated by 0.10mg L$^{-1}$ in
general. In addition, the temporal patterns of DO averaged over the PRE which is represented as the red box in Fig. 5
are consistently matched with RCA simulations, even with the deviations more or less 0.10mg L$^{-1}$. A linear regression
is shown in Fig. 6b with the regression coefficient R$^2$>0.99 and the regression slope lying close to 1:1 ratio line.

27       Despite the overall good agreement between the modulation and RCA simulation, we now focus on the diagnostic

comparisons between the modulation and RCA simulations in terms of the magnitude and contribution of each
individual processes, including horizontal advection, vertical advection, and vertical diffusion (see Fig. 6 c-e). The
horizontal diffusion is much smaller than the above terms and hence neglected. The agreement indicates that the
modulation method is also reasonable for use in the diagnostic analysis.
**4.   Results and discussion**
4.1 Characteristics of DO distribution during the summer of PRE

34       The spatial distribution of DO averaged over two months from July to August in the surface and bottom is shown



in Fig. 7a, b. Compared with the bottom layer, DO concentration in the surface is higher in most of areas except in the
upper estuary (see Fig. 7a) which receives a large number of low oxygen water (DO=4mg $L^{-1}$) discharged from the
river outlets. In the bottom, the lowest DO concentration is about 2mg $L^{-1}$, and it appears between the Jitimen and
Modaomen sub-estuary, near the Gaolan Island (see Fig. 7b). There is a slender zone with relatively lower DO
concentration located on the shelf off the Modaomen sub-estuary linking the Gaolan Island and Hengqin Island.
However, the simulated mean DO concentration remains above 3 mg $L^{-1}$, which has long been used as the threshold of
hypoxia in the PRE (Luo et al., 2008). In terms of this, we estimate the hypoxic frequency in each model grids as
follows in order to identify whether the hypoxia has occurred in this zone during the two months.
$$P = \frac{N}{N_s} * 100 \%$$   (15)
Where $N$ is the number of hours when hypoxia occurs, and $N_s$ is the total number of hours for two months (i.e.,
1488). When the hypoxia is defined as DO below 2 mg $L^{-1}$, which is widely used in the study of hypoxia (Rabalais et
al., 2010), the highest hypoxic frequency is approximate 40 % and it occurs in where the lowest DO concentration
locates. When the threshold of hypoxia increases to 3 mg $L^{-1}$, there is a high frequency zone (HFZ) that can be
observed off the Modaomen sub-estuary. The HFZ zone is encompassed by the isoline of 10 % and resembles the low
DO concentration zone. Within the HFZ, hypoxic frequency ranges from 10 % to about 50 %, indicating the HFZ is
most possible to form hypoxia.
Figure 7e, f show the vertical patterns of DO concentration along the two sections which represent the central
areas of the HFZ. Section A starts from Modaomen sub-estuary and extends southward, while section B starts from the
Gaolan Island and ends near the Hengqin Island. The parallel distribution of DO concentration to bottom topography is
observed in both sections during the July and August. In the section A, the surface DO is between 6 and 7 mg $L^{-1}$ and
the lowest bottom DO is as low as 4 mg $L^{-1}$, which occurs in the middle of the section (see Fig. 7e). In addition, the
relatively lower DO is confined to a thin layer above the sediment. In the south end of the section, where the depth is
as deep as 25 m, the surface DO with a concentration of 7 mg $L^{-1}$ can penetrate to deeper than 15 m. The same is true
for section B (see Fig. 7f), where the surface DO is above 6 mg $L^{-1}$ and the bottom DO is 4 mg $L^{-1}$.
When it is compared with the Chesapeake Bay (Hagy et al., 2004) and the northern Gulf of Mexico (Scavia et al.,
2003;Rabouille et al., 2008), which are known for undergoing hypoxia, the hypoxia in the PRE is much less severe
with the comparable higher DO concentration and the lower hypoxic frequency in the bottom, as well as the relatively
more confined extent of hypoxia. In this study, we estimate the hypoxic area as follows:
$$S = \sum P * \Delta s$$   (16)
Where $P$ is the hypoxic frequency calculated by Eq. (15) and $\Delta s$ is the area of each model grid cell. According to the
statistics, $S$ is an expectation of the hypoxic area which takes temporal variability of hypoxic area into consideration.
When we define the threshold of hypoxia as 2 or 3 mg $L^{-1}$, the expected hypoxic area is 67 and 237 $km^2$ respectively,
and is much smaller than that in the Chesapeake Bay and the northern Gulf of Mexico. Based on what have been
discussed above, three questions are raised to be investigated in the remaining of this manuscript. These three
questions are: (1) which processes control the DO balance in the summer of the PRE, (2) which processes cause the





hypoxia most likely to occur off the Modaomen Sub-estuary, (3) and which processes determine the hypoxia is not
severe in the PRE.
4.2 DO balance
In order to investigate which processes control the DO conditions, a diagnostic analysis of DO balance was
conducted for the PRE (see Fig. 8a) and HFZ (see Fig. 9a). Figure 7d shows that the HFZ is located on the shelf off the
Modaomen sub-estuary. It is encompassed by the isoline of 10 % when we define hypoxia as DO<3 mg L$^{-1}$, and covers
an area of 500 km$^2$. In the diagnostic analysis, abbreviation PAR represents localized partial derivatives of DO; SOD
and $DO_{SOD}$ the sediment oxygen demand and the decrease in DO concentration due to the effects of sediment oxygen
demand; WCR and $DO_{WCR}$ the water column respiration and the decrease in DO concentration due to the effects of
water column respiration; Phot and $DO_{Phot}$ the photosynthesis and the increase in DO concentration due to
photosynthesis; Rea and $DO_{Rea}$ the re-aeration and the increase in DO concentration due to re-aeration; $DO_{BC}$ the
simulated DO concentration only forced by boundary conditions; VADV the vertical advection of DO; HADV the
horizontal advection of DO; VDIFF the vertical diffusion of DO; and HDIFF the horizontal diffusion of DO. We have
argued earlier that DO concentration can be separated into the $DO_{BC}$ (the simulated DO concentration only forced by
boundary conditions) and $DO_{Bio}$ (the increase or decrease in DO concentration only due to the effects of
biogeochemical sources and sinks). This implies that we can estimate the contributions of each biogeochemical terms
as well as the boundary conditions to vertical advection (see Fig. 8b and Fig. 9b), vertical diffusion (see Fig. 8c and Fig.
9c), and horizontal advection (see Fig. 8d and Fig. 9d) as Eq. (13). Horizontal diffusion is much smaller than the above
terms and hence is omitted. Fig. 8e and Fig. 9e show the gross contributions of boundary conditions, ambient
biogeochemical processes, and local biogeochemical processes to DO balance for the PRE and HFZ, respectively.
All of these terms are integrated at each desired grid cell and given for the surface layer, middle layer, and bottom
layer. According to the survey data of the PREPP project (Pearl River Estuary Pollution Project)(Chen et al., 2004), the
pycnocline in the PRE is located in the depth ranging from 1.5 to 3 m. We therefore define the surface layer as the top
20 % of depth for simplicity in view of the 10 m averaged depth in the PRE. The bottom layer is limited to 20 % of
depth above the sediment where the DO concentration is relatively lower (as demonstrated in Fig.7e, f) and hypoxia
most occurs.
4.2.1    PRE
In the surface layer, there is a re-aeration flux across the air-sea interface due to the presence of oxygen gradient
between the water and atmosphere. In the summer of the PRE, there is a DO supplement weighing about 9051 t
occurring in the surface layer every day, causing an increase of averaged DO concentration by 0.55 mg L$^{-1}$ in the upper
20 % thickness of the PRE (see Fig. 8a). Although the re-aeration only occurs in the surface layer, the reoxygenation
will make the DO vertical gradient form and be a supplement of DO in the middle and bottom layers through the
vertical diffusion. According to Fig. 8c, the vast majority (89 %) of oxygen which enters the surface layer from the
atmosphere will be transported to the lower layers through the vertical diffusion, and eventually 28 % reach the bottom
(Fig. 8c). That is why the vertical diffusion is a sink of DO concentration in the surface layer. In addition, there also



exists a significant number of the oxygen replenished by the re-aeration involved in the circulation processes, such as
the horizontal and vertical advections. Figure 8c also reveals that re-aeration is a major contributor to the vertical
diffusion which contributes to 99 % of the vertical diffusion flux. Another important source is photosynthesis. Unlike
the re-aeration, photosynthesis occurs in the water body so that the vertical DO gradient is not so large. As a result, the
oxygen generated by photosynthesis rarely reaches the lower layers through vertical diffusion, but will be transported
by circulations including the horizontal and vertical advection (see Fig. 8b, c, d). Figure 8b, d show that an average of
0.56 mg L$^{-1}$ (accounting for 77 % of horizontal advection) oxygen per day is transported off the PRE while at the same
time the vertical advection brings about 0.35 mg L$^{-1}$ (accounting for 42 % of DO flux caused by vertical advection)
oxygen from the middle layer to the surface layer, both of which are 2 and 1.2 times of the photosynthesis in the
surface layer, respectively. In addition to photosynthesis, the boundary condition is also a major contributor to the
horizontal and vertical advections, and its contribution to the DO flux reaches 0.54 mg L$^{-1}$ (accounting for 77 % of
horizontal advection) and 0.77 mg L$^{-1}$ (accounting for 94 % of vertical advection), respectively (see Fig. 8b, d). Water
column respiration is the only biogeochemical sink in the surface layer and it is similar to photosynthesis for its
occurrence in the water body and participation in circulations (see Fig. 8b, d). For sediment oxygen demand, the
traditional views believe that it occurs in the bottom layer and hence its impact on the surface layer will not be
considered. However, sediment oxygen demand will make a decline of the bottom DO concentration, thereby reduce
the upward DO flux reaching the surface layer, and eventually exert a negative effect on DO concentration in the
surface layer (see Fig. 8b). In general, the ambient and local biogeochemical processes are the most important factors
controlling the DO balance. Boundary conditions including river boundaries and open boundaries can affect the DO
concentration in the surface layer through circulations. However, since the horizontal and vertical advections
compensate each other, the net effects of boundary conditions appear limited (see Fig. 8e).
The middle layer is not influenced by re-aeration and sediment oxygen demand directly, therefore, photosynthesis
and water column respiration become the only two biogeochemical processes affecting the DO balance. However,
since the effects of photosynthesis and water column respiration are relatively small, the middle layer is mainly
controlled by the dynamical processes, in which horizontal and vertical advections are major sinks (see Fig. 8a). In the
middle layer, the oxygen is transported off the PRE by the horizontal advection and upward by the vertical advection.
As the DO content delivered from the middle layer to the surface layer exceeds that transported to the middle layer
from the bottom layer, the overall performance of the vertical advection makes DO concentration decreased. In general,
the DO balance in the middle layer is mainly controlled by boundary conditions.
In the bottom, the magnitude of horizontal and vertical advection decreases sharply since the velocities decrease
(see Fig. 8a). Therefore, unlike the surface layer, the vertical diffusion becomes the major physical source of DO in the
bottom layer. From the perspective of biogeochemical processes, since light weakens in the bottom, the growth of
phytoplankton is limited along the narrow coastal areas. As a result, the photosynthesis and water column respiration
play the trival roles in DO balance and sediment oxygen demand therefore becomes the major biogeochemical sink of
DO in the bottom layer. This is in contrast to the situations in the Chesapeake Bay (Li et al., 2015). The discrepancies
in the dominant role of sediment oxygen demand responsible for the total oxygen depletion in the bottom results from
the differences in geometry. The Chesapeake Bay has a relatively deep channel where the sediment oxygen demand



ranges from 0.86 to 3.2 g m$^{-2}$ day$^{-1}$ , and the sediment oxygen demand only accounts for 16 % of total DO depletion in
the bottom layer(Boynton and Kemp, 1985). However, the PRE is characterized by the shallow banks less than 5m
where the sediment oxygen demand ranges from 0.49 to 3.5 g m$^{-2}$ day$^{-1}$ and causes a decline of 0.53 mg L$^{-1}$ day$^{-1}$ of
DO in the bottom 20 % thickness. Although the sediment oxygen demand only occurs in the bottom layer. However,
this deoxygenation will result in the vertical DO gradient between the middle and bottom layers and facilitate the
oxygen in the middle layer supplement the bottom layer through the vertical diffusion. This process can be viewed as
the effects of sediment oxygen demand is released from the bottom layer, then passes to the middle layer through the
vertical diffusion, and consequently leads to a decrease of oxygen consumptions in the bottom layer and an increase in
the middle layer. Figure 8c shows the sediment oxygen demand is a major cause of vertical diffusion followed by
re-aeration, both of which contribute 79 % and 25 % of vertical diffusion flux, respectively. It should be noted that the
vertical advection can also bring the effects of sediment oxygen demand to the middle and surface layer. In fact, the
vertical advection is the major mechanism of the effects of sediment oxygen demand to reach upper layers, while the
vertical diffusion can only bring the effects to a thin layer above the sediment (see Fig. 8b, c). Besides the vertical
diffusion and sediment oxygen demand, circulations including horizontal and vertical advections control the DO
balance to some extend as well (see Fig. 8a). In the PRE, the light freshwater spreads seaward in the surface layer
accompanied by an upward mixing of the heavy saline water, and consequently a landward compensation flow occurs
in the bottom layer (Dong et al., 2004). This circulation is characterized as a two-layer model and therefore explains
the seaward DO transport in the surface while the landward DO transport in the bottom, as well as the upward DO
transport in the vertical directions. In general, since the horizontal and vertical advections balance each other, the DO
balance in the bottom layer is mainly controlled by the ambient and local biogeochemical processes (see Fig. 8e).
4.2.2    HFZ
When compared with the surface layer in the PRE, the DO flux of vertical advection in the HFZ is almost 5.5
folds larger since its shallower depth and larger vertical velocities. As a result, the vertical advection becomes the most
important process controlling the DO balance in the surface (see Fig. 9a). Besides vertical advection, re-aeration is
another important source of DO. In the HFZ, re-aeration brings about 1.24mg L$^{-1}$ oxygen to the surface layer everyday
(see Fig. 9a). However, as well as in the PRE, most (64 %) of the oxygen replenished by re-aeration reaches lower
layers through the vertical diffusion (see Fig. 9c)., and also participates in circulations (see Fig. 9b, d). In addition, the
magnitude of vertical advection even exceeds the vertical diffusion because of the large vertical velocities (see Fig. 9b,
c).
In the middle layer, the DO budget is mainly balanced by horizontal and vertical advections (see Fig. 9a), both of
which are mainly contributed by boundary conditions (see Fig. 9b, d).
As well as in the PRE, the sediment oxygen demand consumes DO in the bottom layer of HFZ and causes a
vertical diffusion flux. Figure 9b, c reveals that the sediment oxygen demand accounts for 88 % of the vertical
diffusion and can affect the upper layers through the vertical advection. The much larger (about 4.3 folds larger than in
the bottom of PRE) magnitude of vertical advection is also observed in the bottom of HFZ (see Fig. 9a). The horizontal
advection brings the oxygen from the boundary conditions to the bottom layer, while the vertical advection transports





the oxygen upward. Since these two processes compensate each other, the DO balance in the bottom layer is mainly
controlled by the ambient and local biogeochemical processes. Generally, the physical and biogeochemical processes
concerning the DO conditions in the bottom layer of the HFZ and PRE are basically similar in terms of DO balance.
However, taking the hypoxia in the HFZ into consideration, it can be referred that only DO balance is not sufficient to
explain the formation of hypoxia and further discussions are still needed.
4.3  Why the hypoxia occurs in the HFZ
A study conducted by Zhang and Li (2010) shows hypoxia in the PRE is associated with the distribution of
sediment oxygen demand. Because of its dominant roles in bottom depletion, hypoxia occurs where the high rate of
sediment oxygen demand is. These spatial distributions of sediment oxygen demand are caused by the front, which
accelerates the deposition of particulate organic matter in the estuary and Modaomen sub-estuary, and hence forms two
distinct areas characterized by the high level of sediment oxygen demand (see Fig. 10a). However, as shown in Fig.
10a, the HFZ is not located in where the maximum of sediment oxygen demand is. Taking this in to consideration, one
possible reason is that other biogeochemical processes such as water column respiration and photosynthesis are not
considered.
In fact, it is hard to estimate the DO consumption rate in the bottom layer since we cannot determine the thickness
affected by sediment oxygen demand. In the Chesapeake Bay (Hong and Shen, 2013;Shen et al., 2013) and northern
Gulf of Mexico (Yu et al., 2015), sediment oxygen demand is divided by thickness of the lower layer in order to gain
its contributions to the DO consumption rate, assuming that the lower oxygen water could mix evenly below the
pycnocline. In this study, since we have argued earlier that the shallow depth and large vertical velocities enable the
effects of sediment oxygen demand reach the surface layer (see Fig. 8b and Fig.9b), we therefore add sediment oxygen
demand which is divided by the depth to water column respiration and photosynthesis to represent the gross depletion
rate, even though the estimations will be underestimated. The negative values represent DO consuming while the
positive values represent DO replenishing. Figure 10b shows the highest gross depletion rate exceeds 1.0 mg L$^{-1}$ day$^{-1}$
and occurs along the coast of PRE. In the estuary, the gross depletion rate ranges from 0.6 to more than 1.0 mg L$^{-1}$
day$^{-1}$, sharply decreased near the mouth of the estuary. However, there is no hypoxia observed in the estuary. In the
bottom of HFZ, gross depletion decreased dramatically from the 1.0 mg L$^{-1}$ day$^{-1}$ near the Modaomen sub-estuary to
less than 0.2 mg L$^{-1}$ day$^{-1}$. This indicates that only the biogeochemical processes are not sufficient to explain the
hypoxia in the HFZ since the effects of physical processes are also important. However, it remains difficult to quantify
the effects of physical processes on the distribution of hypoxia.
The proceeding DO balance analysis suggests that bottom DO is mainly controlled by ambient and local
biogeochemical processes. Although the boundary conditions can exert an impact on DO concentration through
circulations, the horizontal and vertical advections balanced each other, making the whole effects ignorable. Therefore,
we will focus on how ambient and local biogeochemical processes affect DO distributions. First, the $DO_{Bio}$
distribution in the bottom is examined since it is determined only by these two processes. Figure 10c shows the
simulated $DO_{Bio}$ distribution in the bottom is in reasonable agreement with DO distribution, and the lowest $DO_{Bio}$
concentration mainly occupies the HFZ. This means in the HFZ, biogeochemical processes exert largest negative



effects and hence decrease the DO concentration most violently.
Next, in order to further investigate why the biogeochemical processes have the largest negative effects in the HFZ,
the spatial distributions of $DO_{SOD}$, $DO_{WCR}$, $DO_{Rea}$, and $DO_{Phot}$ are studied (see Fig. 11). In the bottom, the lowest
$DO_{SOD}$ is observed mainly in the coastal areas extending from the west of lower estuary to the HFZ, with a value
ranging from 4 to 5 mg L$^{-1}$ (see Fig. 11a). Since sediment oxygen demand is the most important biogeochemical sink
of oxygen in the bottom, these coastal areas are most conductive to the formation of hypoxia. Reasons that no hypoxia
occurs in the west of lower estuary is related to re-aeration. According to the model results, there are high oxygen
influxes from the atmosphere in the upper estuary (not displayed). Because of the physical modulation, in the bottom
layer, re-aeration has the strongest positive effects on the west of lower estuary (see Fig. 11c). In general, in the west of
lower estuary, the high sediment oxygen demand rate is compensated by the high re-aeration replenishing rate, while in
the HFZ, sediment oxygen demand dominates the DO changes and exerts a strong negative effect on the DO
concentration. As a result, the HFZ is most conductive to hypoxia. Figure 11 also illustrates the reasons for that there is
no hypoxia in the upper estuary in spite of its high rate of DO consumption rate. This is because the upper estuary is
adjacent to the river network and hence is influenced by river discharges largely. As a result, the quick water exchange
brings low DO water parcels out of the bottom layer of upper estuary quickly, making the hypoxia not easy to happen.
This can be demonstrated by low concentrations of $DO_{SOD}$ and $DO_{WCR}$ in Fig. 11a, b.
Finally, in order to investigate why re-aeration has strongest positive effects on the west of lower estuary, the
budget of $DO_{Rea}$ in this area is conducted (see Fig. 11 e-g). The area is encompassed by the isoline of 4mg L$^{-1}$ (black
lines in Fig. 11c). Fig. 11e shows a re-aeration flux across the air-sea interface, among which the vast majorities (76 %)
are transported to the lower layers (see Fig. 11e) and eventually 21 % reach the bottom (see Fig. 11g). In addition,
thanks to the high re-aeration rate in the surface of upper estuary, oxygen in the bottom is also fueled by the horizontal
advection, which brings about 0.31mg L$^{-1}$ oxygen every day from the upstream (see Fig. 11g). Since the supplement
brought by vertical diffusion and horizontal advection exceed the loss caused by vertical advection, there remains a
considerable amount of oxygen replenish by re-aeration in the surface. In the HFZ, the re-aeration flux is 0.39 times
lower and hence the amount of oxygen reaching the bottom layer through vertical diffusion is only one fourth of that in
the west of lower estuary. When compared with turning off the photosynthesis and water column respiration (see
Fig.9921a), turning off the re-aeration (see Fig.9921b) leads to a more significant expansion of hypoxic area and
results in a shift of hypoxic center to the west of lower estuary.
4.4 Why the hypoxia in the PRE is not severe
Unlike extensive hypoxia which exists in the Chesapeake Bay (Hagy et al., 2004) and northern Gulf of Mexico
(Rabouille et al., 2008), hypoxia in the PRE is more like confined to small areas. According to the results of previous
studies, it is related to the dynamical conditions in the PRE. When compared with the long residence time in
Mississippi river (more than 95 days, Rabouille et al. (2008)), the short residence time (3-5 days) in the PRE prevents
the organic matter from completing their biogeochemical cycling (Yin et al., 2004). Moreover, the phosphorus
limitation and high turbidity also inhibit the complete utilization of nutrients and growth of phytoplankton in the PRE
(Yin et al., 2004). However, these explanations are not convincing enough. According to a study conducted by Zhang





and Li (2010), hypoxia in the PRE is mainly controlled by sediment oxygen demand and the sediment oxygen demand
is more related to terrestrial input of particulate organic matter. Therefore nutrients and the growth of phytoplankton
should have few effects on hypoxia in the PRE.
Nevertheless, in this study, we compare the gross depletion rate in the PRE with that in the northern Gulf of
Mexico and Chesapeake Bay (see Table 3). The gross depletion rate is computed as the sum of sediment oxygen
demand and net water column respiration. As shown in Table 3, in the northern Gulf of Mexico, sediment oxygen
demand ranges from 0.06 to 0.70 g m$^{-2}$ day$^{-1}$ during the summer of 2003-2006. Below pycnocline, the net water
column respiration which includes water column respiration and photosynthesis ranges from 0.57 to 3.60 g m$^{-2}$ day$^{-1}$.
Therefore, the gross depletion rate ranges from 0.11 to 0.55 mg L$^{-1}$ day$^{-1}$, with the areal extent of hypoxia averaged
13,500 km$^2$. In the summer of the Chesapeake Bay, the gross depletion rate ranges from 0.16 to 0.96 mg L$^{-1}$ day$^{-1}$ in
the mainstem of the bay, where the persistent hypoxia extends for 8 km$^3$. While in the PRE, the model results (see Fig.
10b) show the gross depletion ranges from less than 0.2 to more than 1.0 mg L$^{-1}$ day$^{-1}$ with the spatial average of 0.47
mg L$^{-1}$ day$^{-1}$ in the estuary and 0.40 mg L$^{-1}$ day$^{-1}$ in the HFZ. Therefore, in terms of the relatively high gross depletion
rate and confined hypoxic area (237 km$^2$), neither the high concentration of sediment, nor phosphorus limitation can be
convincing enough to explain why the hypoxia in the PRE is not severe.
Hypoxia in the PRE is not severe in terms of two aspects, including the limited hypoxic extent and relatively high
DO concentration. In this study, we conduct the correlation analysis of bottom DO against $DO_{Bio}$ and $DO_{BC}$
concentration (not displayed). The strong (R$^2$>0.9) and significant (Sig. <0.01) relations between DO and $DO_{Bio}$
concentration confirms that bottom DO concentration is more associated with $DO_{Bio}$ than $DO_{BC}$ concentration,
which indicates the $DO_{Bio}$ concentration can be used to interpret hypoxia in the PRE. Figure 10c shows the simulated
$DO_{Bio}$ distribution reproduces the spatial distributions of hypoxia and DO concentration in the bottom. We have
argued earlier that the HFZ forms due to the physical modulation of biogeochemical processes, especially sediment
oxygen demand and re-aeration, making the largest negative effects occurs in the HFZ. According to Fig. 10c, the
$DO_{Bio}$ concentration ranges from -2 to -3 mg L$^{-1}$ in the bottom of HFZ, and DO concentration hence ranges from 3 to
4 mg L$^{-1}$ considering the $DO_{BC}$ concentration is more or less 6 mg L$^{-1}$. Therefore, the high DO concentration can be
attributed to the relatively low effects of biogeochemical processes.
It is related to re-aeration that $DO_{Bio}$ concentration is generally low. When compared with the Chesapeake Bay
the northern Gulf of Mexico, the PRE is characterized with relatively high sediment oxygen demand and shallow depth.
Therefore, the impact of sediment oxygen demand on the bottom DO is comparable more important. Figure 13 reveals
$DO_{SOD}$ ranges from -4 to -5 mg L$^{-1}$ in the bottom of HFZ, with the spatial averaged value of -4.31 mg L$^{-1}$. This
indicates the averaged DO concentration in the bottom of the HFZ will be as low as 1.76 mg L$^{-1}$ and the expected
hypoxic area will reach 3345 km$^2$, presuming that other biogeochemical processes are neglected. Figure 13 also reveals
that photosynthesis offsets the water column respiration and eventually supplements the DO concentration by 0.98 mg
L$^{-1}$ in the bottom. Re-aeration is another important source which averages 0.88 mg L$^{-1}$ in the bottom. According to Fig.
12, taking either re-aeration or photosynthesis and water column respiration into consideration leads to the hypoxic
area decrease to 591 km$^2$ and 2203 km$^2$, respectively. Moreover, Fig. 12 also reveals that without re-aeration, the west
of lower estuary is occupied with a persistent hypoxia (frequency>80 %), noting the fact that hypoxia is intermittent in



1    the PRE (Zhang and Li, 2010).

## 5.    Summary and conclusions

In this study, we use a physical and biogeochemical coupled model to investigate the DO dynamics and hypoxia during summer in the PRE. Comparisons with observations demonstrate that our model reasonably reproduces the observed spatial and temporal characteristics of water level, salinity, temperature, and DO. The good agreement between the model simulated and historical estimated rates of re-aeration, sediment oxygen demand, and water column respiration further indicates our model can accurately simulate the biogeochemical processes concerning the DO dynamics. In addition, we introduce a novel method named physical modulation of biogeochemical terms to investigate the contributions of boundary conditions, ambient biogeochemical processes, and local biogeochemical processes to DO conditions. The formula derivation and comparisons against model outputs reveal the modulation method is reasonable for use in DO analysis.

Model results demonstrate there is a high frequency zone (HFZ) of hypoxia located on the shelf off the Modaomen sub-estuary. However, when compared with other areas, hypoxia in the PRE is not severe in terms of its intermittency and limited extent. Based on the modulation method, a diagnostic analysis of DO balance is conducted for the PRE and HFZ to bring us a further insight into DO dynamics. The analysis results show that the bottom DO conditions are mainly controlled by the ambient and local biogeochemical processes, both in the HFZ and PRE. Although the circulation process can bring the DO originating from the boundaries to the bottom of the PRE and HFZ, the influx of horizontal advection and outflux of vertical advection compensate each other, and hence the total impacts of boundary conditions are limited.

Since $DO_{\mathrm{Bio}}$ concentration is determined only by ambient and local biogeochemical processes, we compare $DO_{\mathrm{Bio}}$ and DO in terms of their spatial and temporal distributions. A good agreement further indicates that $DO_{\mathrm{Bio}}$ can be used to interpret formations of hypoxia. Re-aeration and sediment oxygen demand are two main biogeochemical processes which control the distribution, the spatial extent, and the duration of hypoxia in the PRE. Though the high rate gross depletion in the upper of the estuary, the hypoxic water in the bottom is soon diluted because of quick water exchange. In the HFZ and the west of lower estuary, sediment oxygen demand decrease bottom DO concentration distinctly, making these two areas potentially hypoxic. However, oxygen entering the surface layer through the re-aeration will be transported to the bottom in the west of lower estuary, offseting the consumed oxygen by sediment oxygen demand and therefore eliminating the hypoxia. Since this mechanism is not distinct in the HFZ, HFZ becomes the most likely to form hypoxia in the PRE. Numerical simulations reveals that turning the re-aeration leads to a northward expansion of hypoxic extent to the west of lower estuary with the persistent hypoxia observed.

## Acknowledgements

This work was supported by the National Natural Science Foundation of China (Grant No: 41306105), the Guangdong Natural Science Foundation (Grant No: 2014A030313169), the Science and Technology Planning Project of Guangdong Province, China (Grant No: 2014A020217003), and the Fundamental Research Funds for the Central Universities (Grant No: 131gpy59).





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





Table 1. Parameters for the water quality model

| parameters | range | value | units |
|---|---|---|---|
| Maximum specific growth rate of phytoplankton | 1.7-3.0[a] | 2.0[c] | day[-1] |
| Respiration rate | 0.1-0.3 [a] | 0.1 [c] | day[-1] |
| RDON mineralization rate | 0.008-0.01 [a] | 0.009 [c] | day[-1] |
| LDON mineralization rate | 0.085-0.1 [a] | 0.09 [c] | day[-1] |
| RPON hydrolysis rate | 0.007-0.01 [a] | 0.008 [c] | day[-1] |
| LPON hydrolysis rate | 0.05-0.07 [a] | 0.06 [c] | day[-1] |
| RDOP mineralization rate | 0.01-0.02 [a] | 0.01 [c] | day[-1] |
| LDOP mineralization rate | 0.1-0.2 [a] | 0.1 [c] | day[-1] |
| RPOP hydrolysis rate | 0.007-0.1 [a] | 0.008 [c] | day[-1] |
| LPOP hydrolysis rate | 0.085-0.1 [a] | 0.09 [c] | day[-1] |
| RDOC mineralization rate | 0.007-0.01 [a] | 0.009 [c] | day[-1] |
| LDOC mineralization rate | 0.1-0.15 [a] | 0.1 [c] | day[-1] |
| RPOC hydrolysis rate | 0.007-0.01 [a] | 0.01 [c] | day[-1] |
| LPOC hydrolysis rate | 0.07-0.1 [a] | 0.08 [c] | day[-1] |
| RPON,LPON,RPOP,LPOP,RPOC,LPOC settling rate | 0.5-1.0 [a] | 0.5 [c] | m day[-1] |
| Nitrification rate | 0.05-0.1 [a] | 0.08 [c] | day[-1] |
| Denitrification rate | 0.05-0.4 [a] | 0.09 [c] | day[-1] |
| G1 POM diagenesis rate | 0.035[b] | 0.035[b] | day[-1] |
| G2 POM diagenesis rate | 0.0018[b] | 0.0018[b] | day[-1] |
| G3 POM diagenesis rate | 0-1.0E-6[b] | 1.0E-7[c] | day[-1] |
| Nitrification rate in the sediment | 0.1313[b] | 0.1313[b] | m day[-1] |
| Denitrification rate in the aerobic layer | 0.2-1.25 [b] | 1.25 [c] | m day[-1] |
| Denitrification rate in the anaerobic layer | 0.25 [b] | 0.25 [b] | m day[-1] |

[a] HydroQual (2004)
[b] Ditoro (2001)
[c] Zhang and Li (2010)



Table 2 Comparisons between the simulated and historical estimated re-aeration, sediment oxygen demand, and
respiration by phytoplankton in the summer of the Lingdingyang Bay

|  | Simulated | Historical estimated | period |
|---|---|---|---|
| Rea: g m$^{-2}$ day$^{-1}$ | -0.09-9.59 | -0.68-6.8[a] | August 2005; August 2008 |
| SOD: g m$^{-2}$ day$^{-1}$ | 1.01-3.53 | 0.72-3.89[b] | July 1999 |
| Resp: mg L$^{-1}$ day$^{-1}$ | 0.00-0.27 | 0.11-0.37[a] | August 2008 |

[a] He et al. (2014)
[b] Chen et al. (2004)
Table 3 Summary of sediment oxygen demand (SOD), net water respiration (NWCR=WCR+Phot), and gross depletion
rate below the pycnocline reported for the northern Gulf of Mexico and Chesapeake Bay. In addition, the thickness of
the lower layer below the pycnocline and hypoxic area are included.

|  | Period | SOD g m$^{-2}$ day$^{-1}$ | NWCR g m$^{-2}$ day$^{-1}$ | gross depletion mg L$^{-1}$ day$^{-1}$ | Hypoxic area |
|---|---|---|---|---|---|
| **The northern Gulf of Mexico** | Jun. 2003[a] | 0.06-0.70 | 0.57-2.39 | 0.11-0.24 | 13,500 km$^2$ [b] |
|  | Jun. 2006[a] | 0.06-0.58 | 2.69-3.50 | 0.24-0.33 |  |
|  | Aug. 2007[a] | 0.06-0.53 | 1.06-2.23 | 0.23-0.55 |  |
| **The Chesapeake Bay** | August[c] | 1.5-3.2 | 1.7-16.0 | 0.16-0.96 | 8 km$^3$ [d] |

[a] Murrell et al. (2011)
[b] Rabalais et al. (2007)
[c] Boynton and Kemp (1985)
[d] Rabalais et al. (2010)

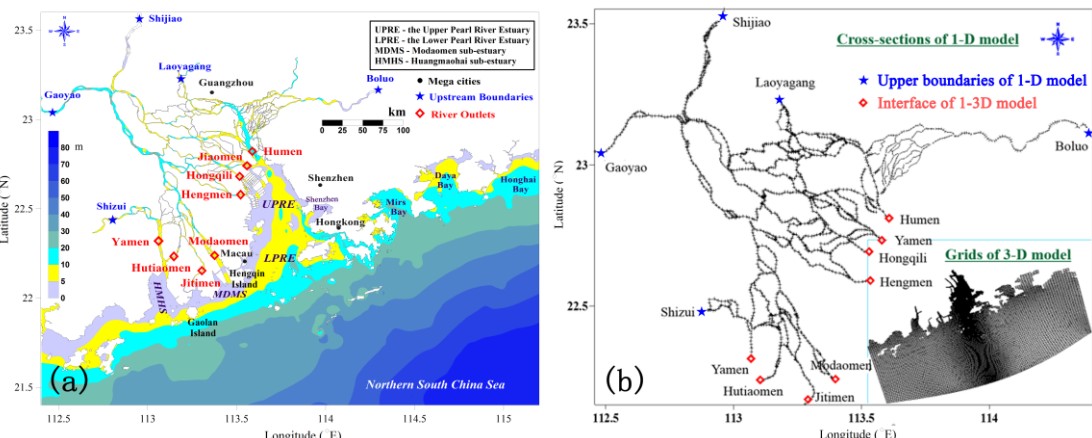

Fig. 1 Maps showing (a) the Pearl River Delta with the Pearl River network and the Pearl River Estuary, and (b)
cross-sections for 1-D model and model grids for 3-D model.

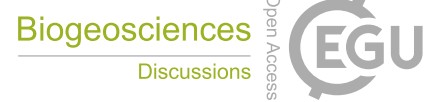



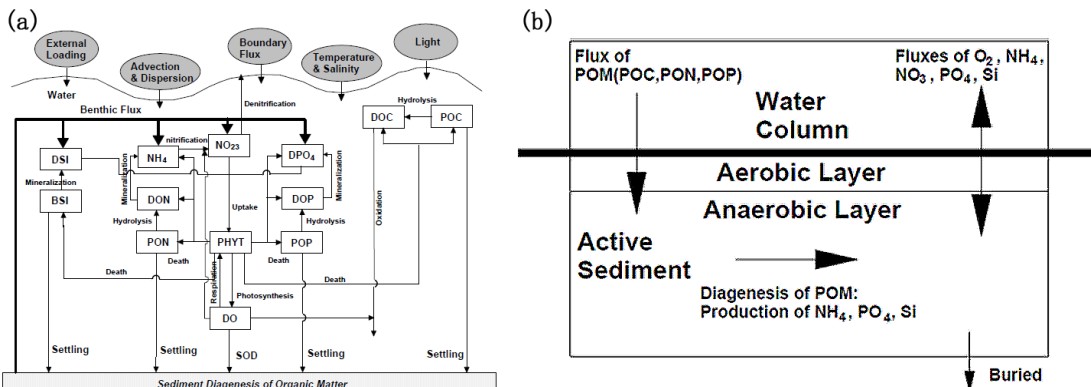

Fig. 2 Conceptual framework for (a) RCA model and (b) sediment flux model (revised from (Ditoro, 2001)). DO represents dissolved oxygen; PHYT represents phytoplankton; POC represents particulate organic carbon; DOC represents dissolved organic carbon; NH$_4$ represents ammonia nitrogen; NO$_{23}$ represents nitrite and nitrate nitrogen; PON represents particulate organic nitrogen; DON represents dissolved organic nitrogen; DPO$_4$ represents dissolved inorganic phosphorus; POP represents particulate organic phosphorus; DOP represents dissolved organic phosphorus; DSi represents dissolved silica; BSi represent biogenic silica; and SOD represents sediment oxygen demand.





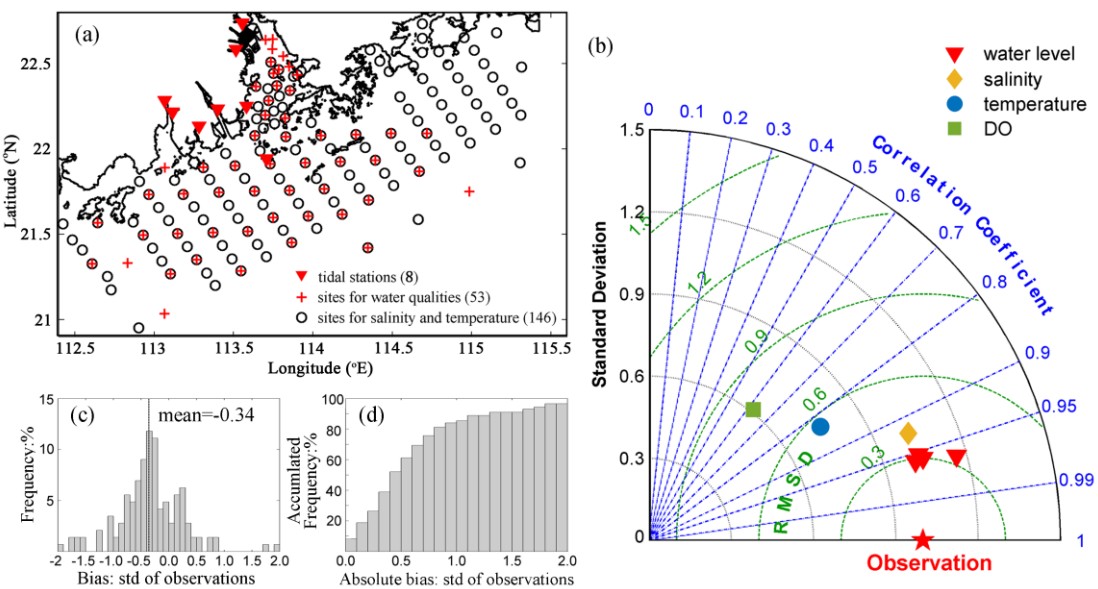

Fig. 3 (a) Survey stations of water levels (red triangles), salinity (black circles), temperature (black circles), and water quality variables (red crosses) during July and August 2006. (b) Normalized Taylor diagram illustrating our model skills of simulating the combined temporal and spatial patterns of water level (red), salinity (orange), temperature (blue), and DO (green). Frequency (c) and accumulated frequency (d) distribution as a function of the biases and absolute values of biases normalized by the standard deviation of observations between the simulated and observed DO during July and August 2006. The positive value means our model overestimates the DO concentration while the negative value means our model underestimates the DO concentration.





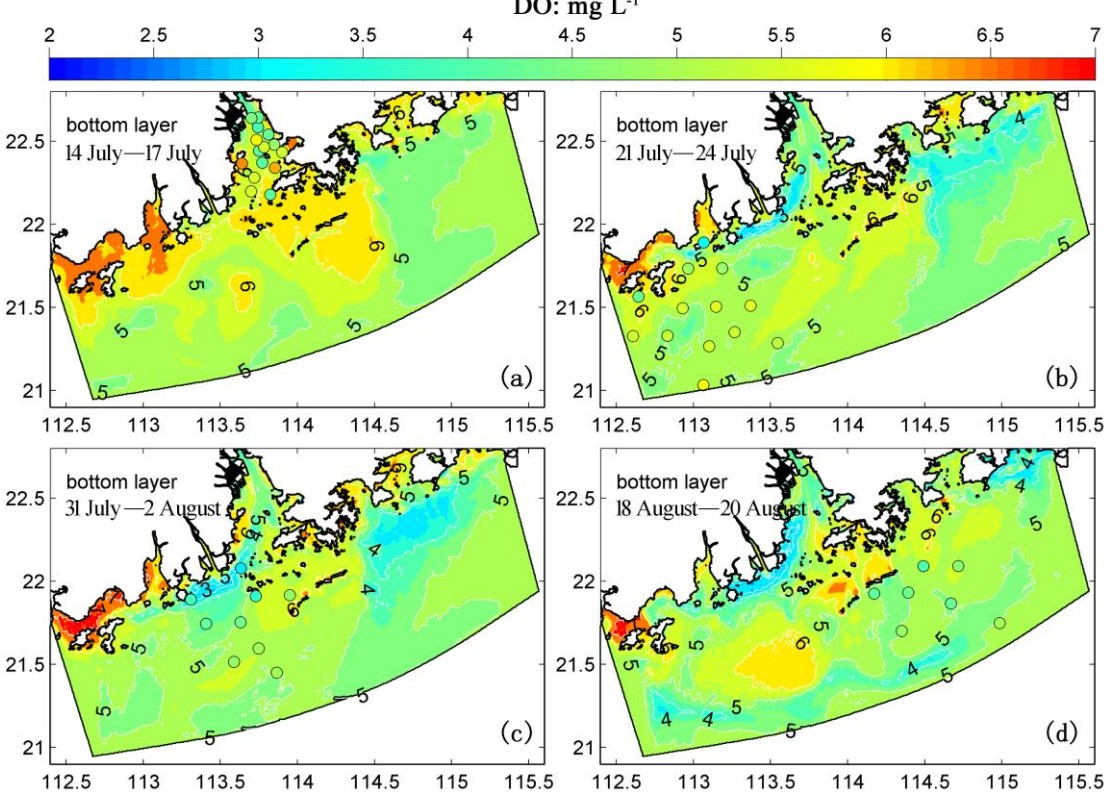

Fig. 4 Simulated (colored map) and observed (colored dots) DO concentration in the bottom layer for July and August 2006.

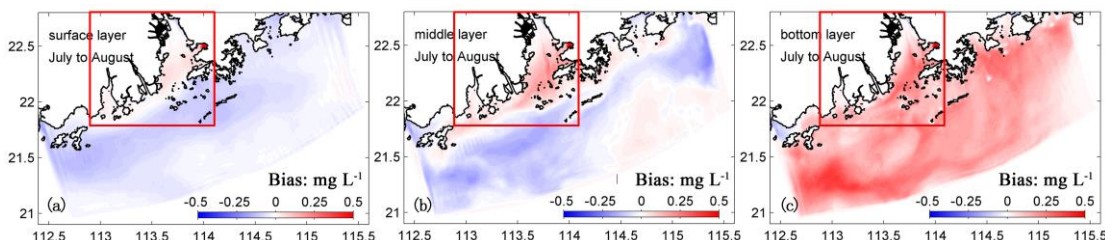

Fig. 5 Spatial distribution of biases between two-month mean (July to August) DO concentration. Positive values (red) indicate modulation method overestimates while negative values indicate modulation method underestimates DO concentration. The red box represents the Pearl River Estuary (PRE) which we focus on in this study.



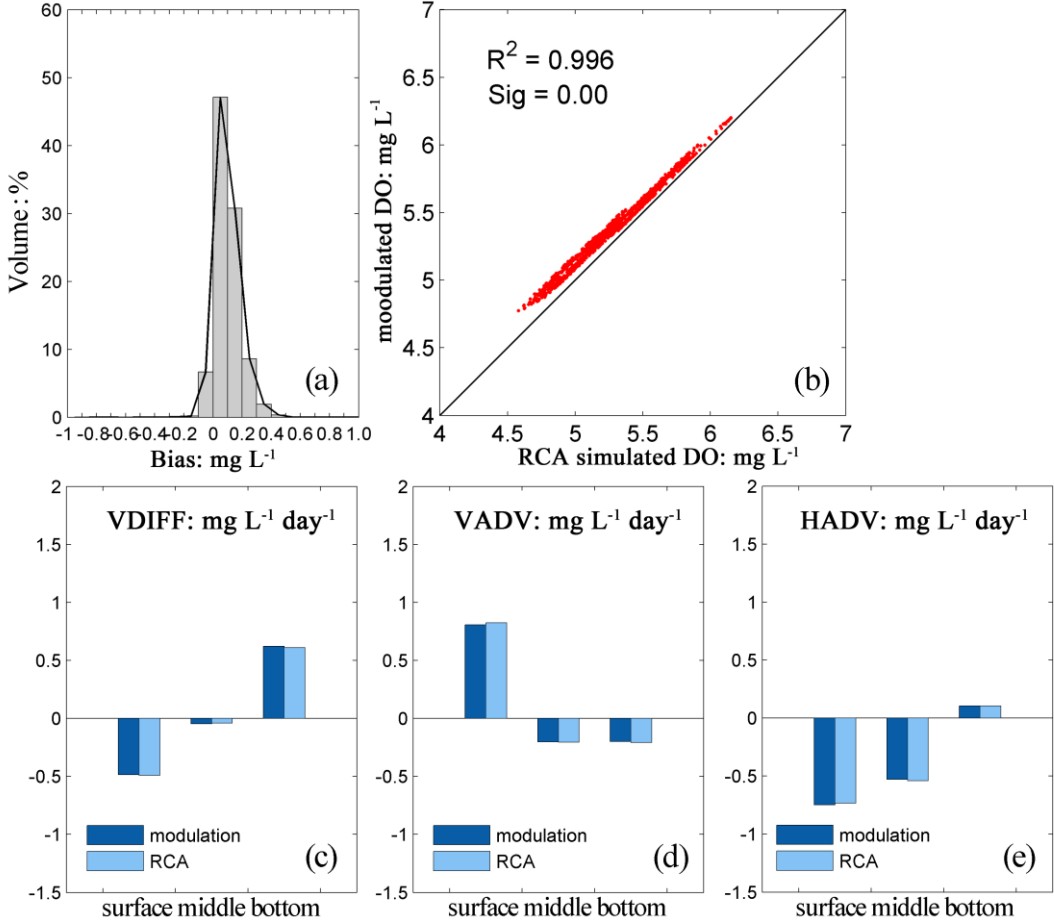

Fig. 6 (a) Volume distribution as a function of biases between the two-month mean DO concentration simulated by the water quality model and modulation method. (b).Scatterplot of spatial averaged DO concentration of the PRE simulated by the water quality model and modulation method. Comparisons of DO fluxes contributed by vertical diffusion (c), vertical advection (d), and horizontal advection (e).





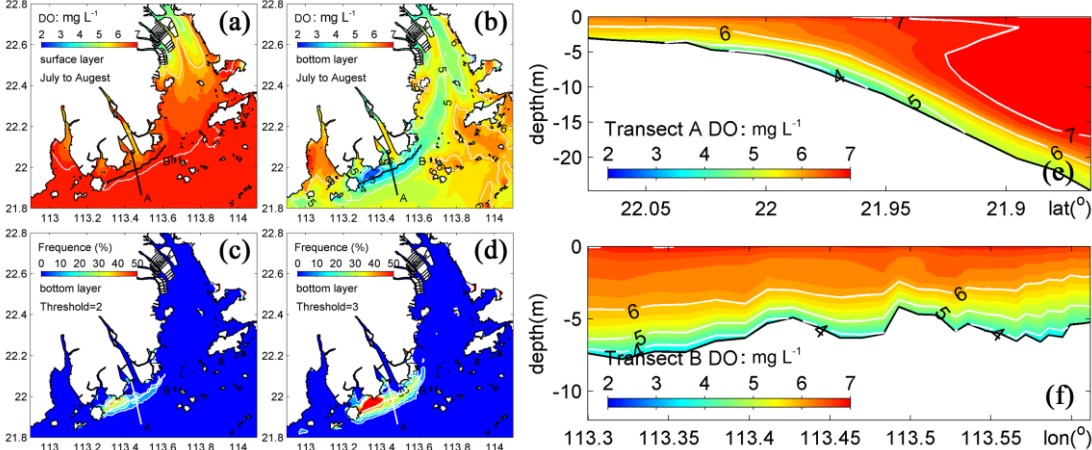

Fig. 7 The spatial distribution of DO averaged over July and August in the PRE in surface (a) and bottom (b); the spatial distribution of hypoxic frequency during July and August in the PRE when hypoxia is defined as DO<2 mg L$^{-1}$ (c) and DO<3 mg L$^{-1}$ (d); the distribution of DO averaged over July and August along the two transects. Positions of the two transects are shown in Fig. 7(a-d).





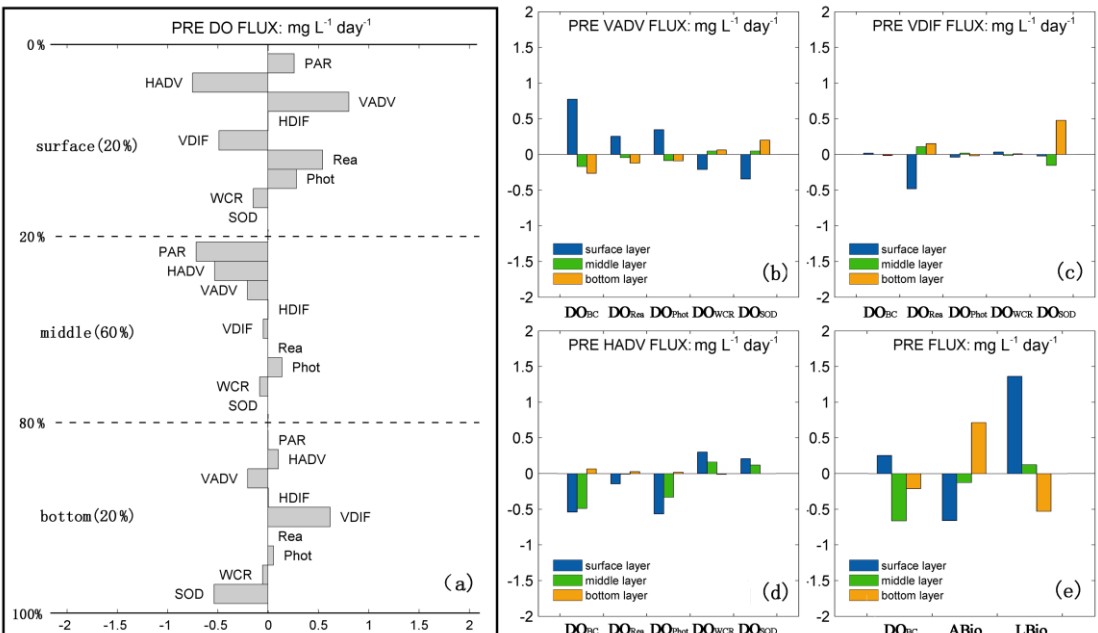

Fig. 8 (a) Simulated two-month (July to August) mean DO balance for PRE. Oxygen physical and biogeochemical terms are given for the surface layer above the 20 % of depth, for the middle layer, and bottom layer which covers 20 % of depth over the sediment. (b) Contributions of each biogeochemical terms as well as boundary conditions to vertical advection flux. (c) Contributions of each biogeochemical terms as well as boundary conditions to vertical diffusion flux. (d) Contributions of each biogeochemical terms as well as boundary conditions to horizontal advection flux. (e) Contributions of boundary condition, ambient biogeochemical process, and local biogeochemical processes to DO balance.





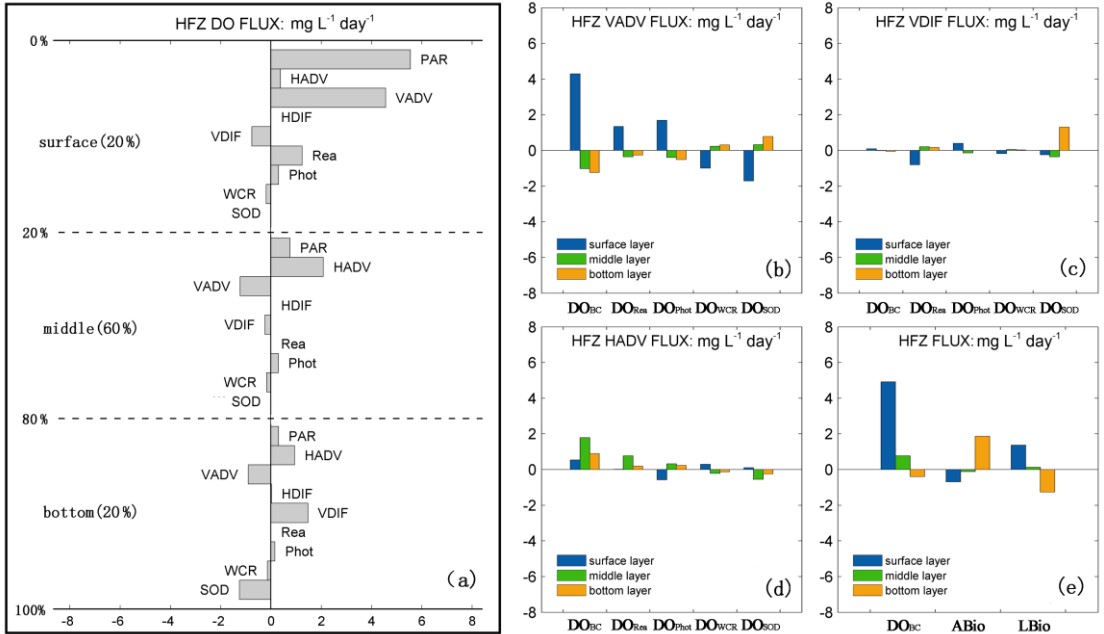

Fig. 9 (a) Simulated two-month (July to August) mean DO balance for the high hypoxic frequency zone (HFZ) which is encompassed by the isoline of 10 % when hypoxia is defined as DO<3mg L$^{-1}$. (b) Contributions of each biogeochemical terms as well as boundary conditions to vertical advection flux. (c) Contributions of each biogeochemical terms as well as boundary conditions to vertical diffusion flux. (d) Contributions of each biogeochemical terms as well as boundary conditions to horizontal advection flux. (e) Contributions of boundary condition, ambient biogeochemical process, and local biogeochemical processes to DO balance.

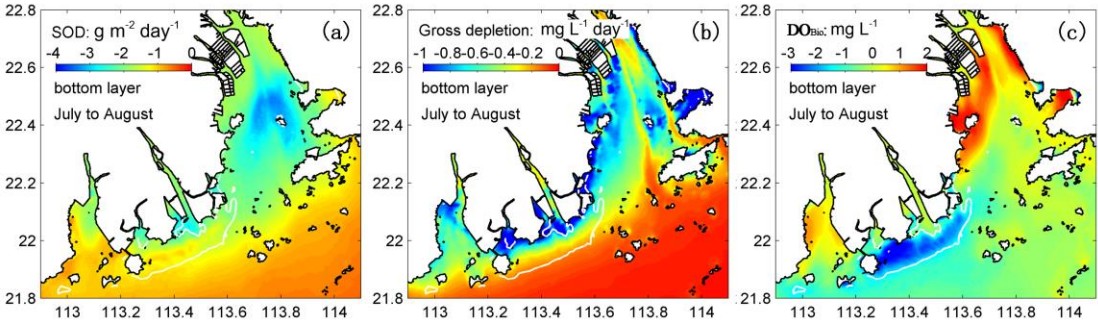

Fig. 10 Distribution of two-month averaged SOD (a), DO gross depletion rates (b), and concentration of $DO_{Bio}$ (c) in the bottom. The HFZ is represented by the area encompassed by white lines.





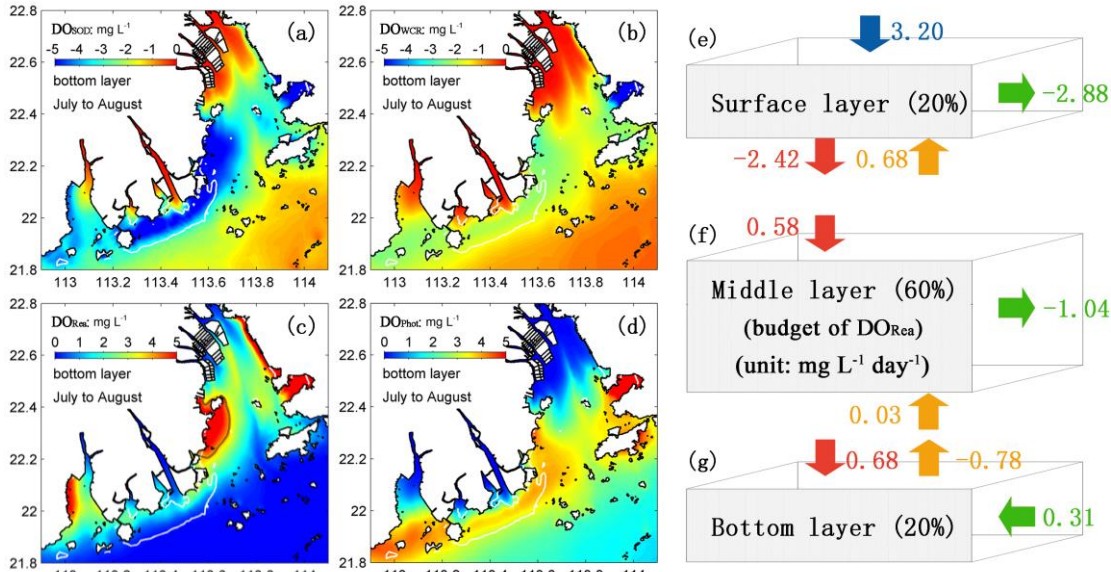

Fig. 11 Distribution of two-month averaged $DO_{SOD}$ (a), $DO_{WCR}$ (b), $DO_{Rea}$ (c), and $DO_{Phot}$ (d) in the bottom. The HFZ is represented by the area encompassed by white lines. Budget of $DO_{Rea}$ in the surface layer (e), middle layer (f), and bottom layer (g) in the west of lower Lingdingyang Bay which is encompassed by black lines in Fig. 11c. The blue arrow represents biogeochemical process (re-aeration), the red arrow vertical diffusion, the orange arrow vertical advection, and the green arrow horizontal arrows. The positive value means a source while the negative value means a sink of DO. (unit: mg L$^{-1}$ day$^{-1}$)





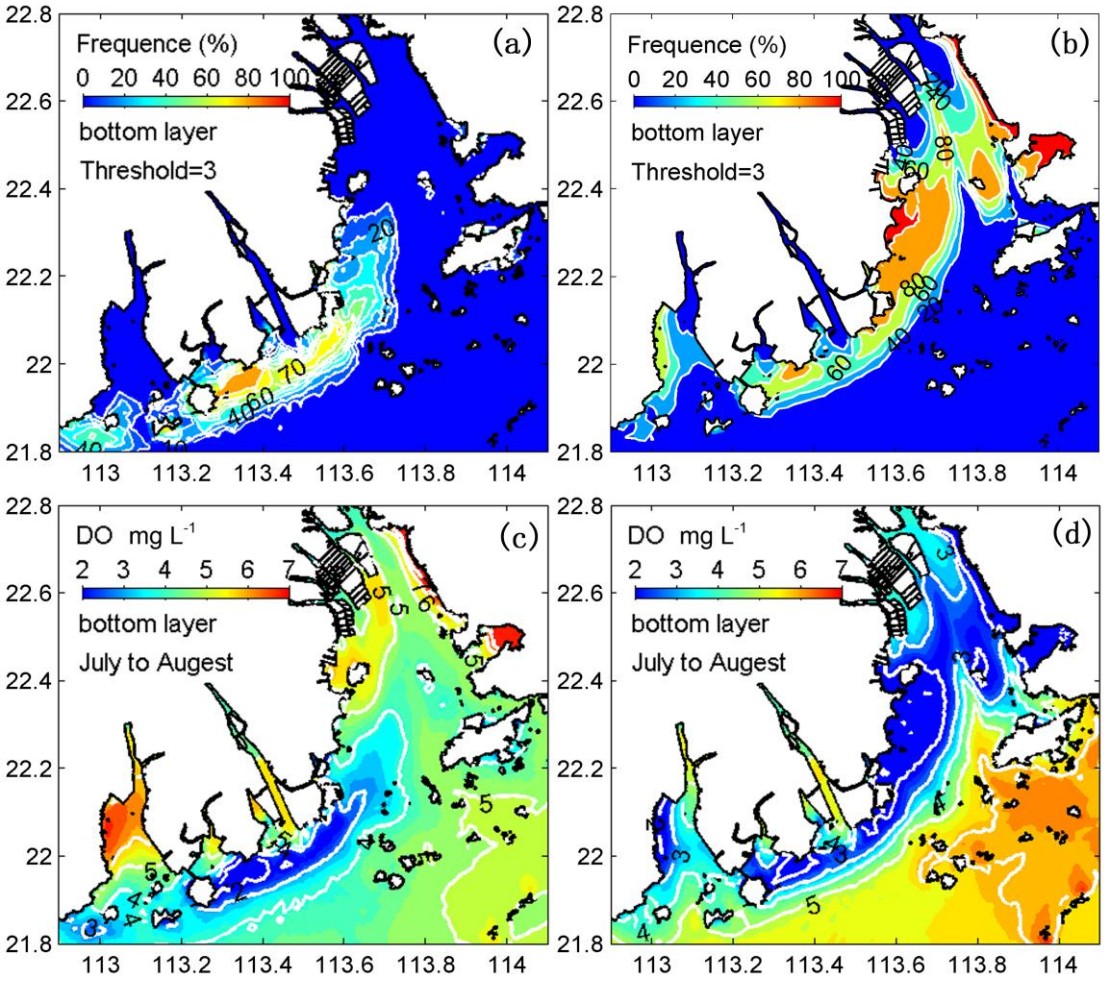

Fig. 12 The spatial distribution of hypoxic frequency and DO concentration during July and August in the PRE when either photosynthesis and water column repiration (a, c) or re-aeration (b, d) are turned off (threshold = 3 mg L$^{-1}$)





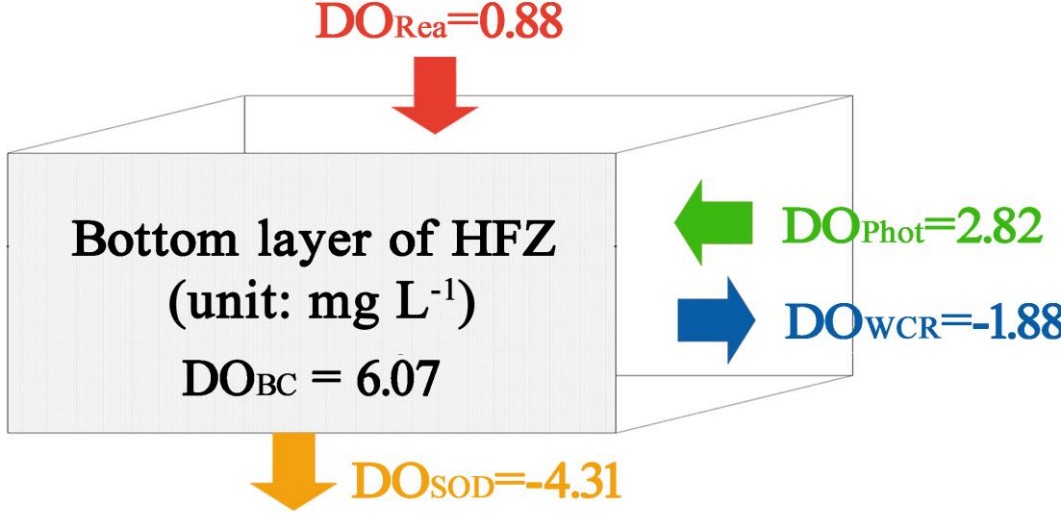

Fig. 13 Balance of DO in the bottom layer in HFZ. The positive value means a positive effect on DO while the negative value means a negative effect on DO (unit: mg L$^{-1}$)