# Peer review of "A numerical analysis of biogeochemical controls with physical"

_Biogeosciences, 2016_

## Referee Comment (RC1) · Anonymous Referee #1 · 19 Dec 2016

Review of "A numerical analysis of biogeochemical controls with physical modulation on hypoxia during summer in the Pearl River Estuary" by Wang et al. This paper reports on the use of a three-dimensional modeling system to explore the biogeochemical and physical mechanisms regulating O2 dynamics and bottom water hypoxia in the Pearl River estuary. A novelty of the study is the use of numerical O2 tracers to quantify the source and sink processes dictating O2 concentrations. The use of these tracers benefitted a diagnostic O2 mass balance in this shallow and dynamic estuary. Further, the study demonstrates the spatial connection between processes occurring in different locations in the system. From this analysis, the conclusion is that air-sea mixing and sediment oxygen demand were the primary processes regulating bottom-water

[Figure]

O2 concentration and hypoxia. This is a different result than other well studied systems where hypoxia occurs such as Chesapeake Bay and northern Gulf of Mexico. Overall, I think this is an interesting paper that for the first time teases apart the biogeochemical and physical aspects of O2 dynamics in a shallow, river-dominated system. Two major issues for the paper are the grammar and imprecise/incorrect use of language. I had to read many sentences two or three times for the meaning to shine through. These issues should be resolved by having a fluent English speaker edit the draft. Specific comments: P1, line 14: The use of 'ambient' here and throughout the paper is confusing to me. Do you mean to use 'adjacent'? It seems like you are inferring the advective and diffusive fluxes from adjacent grid cells. Please clarify. P1, line 25: As you haven't yet defined "modulation of physical processes" this phrase appears to be jargon and is unclear to the reader. P1, line 26: The values presented here and following seem like they should be process rates like 4.31 mg l-1 d-1. Or if this value is an average over some seasonal or annual period, please state this. P1, line 32-33: This sentence doesn't make sense. On the one hand it is stated that photosynthesis increased the O2 concentration, but then it also increased the hypoxic area (decreased O2 concentration). Please clarify. P2, lines 2-7: These sentences are not clear and appear unnecessary. I suggest deleting them and starting the Introduction with line 8 P3, line 4: "Pearl River Estuary", estuary should not be capitalized P3, line 18: Describe here what you mean by "physical modulation of biogeochemical terms". P3-8: Nice description of the modeling P8, line 7: In the 'Model Validation' section you might also suggest some additional observations that could be measured to assess the modulation model. Perhaps oxygen isotopes or additional rate measures would be useful to validate the biogeochemical O2 terms. P11, line 29: unclear what '905 l t' is P12, lines 4-6: I disagree with this statement. The O2 gradient driven by photosynthesis would only be uniform or small if the light were at levels saturating to photosynthesis throughout the water-column P15, line 27: Correct 'Fig.9921a and Fig.9921b P16, line 11: '8 km3', use units consistent with what you present from the Gulf of Mexico and PRE (km2) Table 2: In the heading do you mean 'WCR' instead of 'respiration by phytoplankton',

which is hard to measure in practice and which is usually small in comparison to WCR. Fig. 1b: Did you mean to show the cross-section (side-view) as noted in the figure caption? Here it shows the map view of the river network. Fig. 5: In the caption state the bias is between the two models, RCA and modulation Fig 6: Perhaps I missed it in the text, but why do the two models differ? Please discuss in text. Fig 8e: define ABio and LBio in the caption Technical corrections: There are too many for me to enumerate here. Please have an English editor assist with fixing plural noun/verb issues and other grammatical mistakes.

---

## Referee Comment (RC2) · Anonymous Referee #2 · 27 Jan 2017

This study by Wang et al. present a numerical investigation of the hypoxia condition in one of the largest river estuary along the Western Pacific - the Pearl River Delta. According to the author, the Pearl River has a similar nutrient load as that of the Mississippi and the Pearl river runs through a highly populated region. The importance of this study is beyond question. And the coupling between 1D river model and 3D coastal model as well as the author's efforts of improving the ECOM model is plausible. However, I found this manuscript is not well written and the author failed to explain his/her points throughout the manuscript. The so- called "physical modulation method" is merely to isolate lateral DO from in-situ biological source and sinks. In addition, throughout the manuscript the definition of ambient biogeochemical process is very

vague and sometimes contradictive. The available observation data does not support the high frequency zone (HFZ) proposed by the author. I do agree with the other reviewer that this manuscript needs a throughout edits of scientific writing. I have a lot of problems to understand what the author wants to convey. My detailed comments are as following.

First of all, the abstract contradicts with itself. The author first indicated DO is affected by both local biological processes and DO fluxes from the ambient water (I would treat this as boundary condition). And the author claims the later (ambient water) is unclear. Then in next paragraph, the author indicated that the re-aeration (air-sea, i.e. surface) and sediment oxygen demand (bottom) are most important.

Again, the author indicates the importance of the re-aeration, followed by a statement saying "turning off the re-aeration leads to hypoxia. . ... to the west lower estuary". However, the author immediately states that hypoxia was observed in west lower estuary. So my question is re-aeration good or bad? Important or not important at all?

Then, in the last part of the abstract, the author states photosynthesis is less important. Yet right following this, the author indicates that in HFZ, photosynthesis cause an increase of hypoxia area to 591 km2— it confusing me how photosynthesis would worsen hypoxia (should be respiration, right? But respiration is not directly linked with photosynthesis).

Page 2, line 29-34. The author first indicate DO from ambient water body, followed by " Take the re-aeration as an example". How would these two processes be connected (one is lateral boundary and the other is upper boundary)? The author then mentioned, "given the mechanism remains unclear", which one mechanism? Why it is not clear? Also, I do not think DO flux from ambient water bodies belong to the biogeochemical processes only, it indeed a combination of both bio and physical.

Page 3, line 14-15, the author compared PRE with Northern Gulf of Mexico. I think one of the core mechanisms here is the tidal range difference (M2 in PRE vs S1 in NGOM).

Page 3, line 17. I do not agree that the mechanism remains clear, since just tow lines above the author indicated that previous study suggest sediment oxygen demand and stratification are two main reasons. The author should give a more persuasive justification for this study—I believe there are many indeed.

Page 6. Line 3, what is RDOC, LDOC, ReDOC, ExDOC stands for?

Page 6. Line 15. The authors used a constant boundary condition to study the impact from ambient DO transport—which worries me a lot!

Page 7 Line 7. Should not re-aeration of DO a physical process instead of a biological one?

Page 7. I am confused why the author needs to give repeated information in equation (6)-(12). They simply divide DO in to DObc and DObio, and they can just use equation (13), that is enough.

Page 8. Model validation, I think it is necessary to present more validation for both physics and water quality, specifically some time series comparison. The Taylor diagram could represent the correlation yet we are not sure how the model could resolve temporal variability.

Page 9. Section 3.2. The author presents the difference between RCA and the Physical Modulation method. How about the difference between Physical modulation and the in-situ data? Is there any improvement compared with difference between RCA and in-situ data?

Page 9. Line 30. "The agreement indicates . . ." which agreement?

Page 10. Line 16. The HFZ refers to high frequency zone of areas with DO < 3 mg/L. But why the author states "indicating HFZ is most possible to form hypoxia"?

Page 11. Line 6. What does the author mean by "it is encompassed by the isoline of 10%"?

Page 11. Line 7-16. Should not these abbreviation goes to the figure caption?

Page 13. Line 4. "Although. . .. However . . . "very awkward wording

Page 13, Line 10. "Both of which contribute 79% and 25%.... "I do not understand what the author wants to convey here.

Page 13. Line 19-20. From Figure 8e I could tell how horizontal and vertical advection balance each other.

Page 14. Line 10-11. The author mentioned "two distinct areas characterized by the high level of sediment oxygen demands". However I could not find it on figure 10a referred here (I only see one area delineated by the white line).

Page 14. Line 21-22. I do not understand what does the author mean here "depth to water column respiration and photosynthesis". Or the author means depth of water column respiration/photosynthesis. Also why this estimation is underestimated?

Page 14. Line 27-28. Why such decrease cannot be explained only by biogeochemical processes?

Page 14. Line 34-36. I could not tell how "simulated DO distribution in the bottom is in reasonable agreement with DO distribution". If we look at the available DO observations in Fig.4. There are only 2 points in Figure 4(c) that shows low oxygen in the so-called HFZ zone.

Page 15. Line 2-3. I could not tell that the lowest DOsod is observed in the HFZ zone from Fig 4. Although the author mentioned Fig 11 here, but it is model result rather than in-situ observation.

Page 15. Line 25. Figure 9921a. I do not know what the author what to convey here.

Page 16. Why "phosphorous limitation can not be convincing enough?" The author does not mention much about phosphorous budget before this.

[Figure]

Page 16. Line 20-21. Again, I am not convinced that Fig.10c (model) shows the realistic condition (Fig. 4).

Page 16. Line 28. "is comparable more important". Awkward wording.

Page 17. Line 17. The author used a constant boundary condition, how can he/she justify that "DO originating from the boundaries to the bottom of the PRE and HFZ"?

---

## Author Comment (AC1) · 23 Feb 2017

We wish to thank the referee for the detailed and constructive comments which will be helpful to the revision of our manuscript. According to the comments, we plan to make revisions and improvements as below: 1) to include more results of validations and comparison of model skills between our model and other published studies; 2) to give clearer descriptions and explanations about 'physical modulation method' and its associated concepts; 3) to have an English editor edit our manuscript and correct grammatical mistakes throughout the text.

Detailed responses to all comments are given below. (Responses are shown in *Italic Font*)

**Comment:** This paper reports on the use of a three-dimensional modeling system to explore the biogeochemical and physical mechanisms regulating O2 dynamics and bottom water hypoxia in the Pearl River estuary. A novelty of the study is the use of numerical O2 tracers to quantify the source and sink processes dictating O2 concentrations. The use of these tracers benefitted a diagnostic O2 mass balance in this shallow and dynamic estuary. Further, the study demonstrates the spatial connection between processes occurring in different locations in the system. From this analysis, the conclusion is that air-sea mixing and sediment oxygen demand were the primary processes regulating bottom-water O2 concentration and hypoxia. This is a different result than other well studied systems where hypoxia occurs such as Chesapeake Bay and northern Gulf of Mexico. Overall, I think this is an interesting paper that for the first time teases apart the biogeochemical and physical aspects of O2 dynamics in a shallow, river-dominated system. Two major issues for the paper are the grammar and imprecise/incorrect use of language. I had to read many sentences two or three times for the meaning to shine through. These issues should be resolved by having a fluent English speaker edit the draft.

*Response:*
  *Thank the referee for the positive comments. As suggested, we will have an English editor edit the manuscript.*

**Comment:** The use of 'ambient' here and throughout the paper is confusing to me. Do you mean to use 'adjacent'? It seems like you are inferring the advective and diffusive fluxes from adjacent grid cells. Please clarify.
*Response:*
  *Yes, the contributions of ambient biogeochemical processes represent the advective and diffusive fluxes of oxygen which is produced or consumed by the biogeochemical processes in the adjacent grid cells. According to Eq. 14, the contributions of ambient biogeochemical processes are described as $\Delta t \times [-ADV(DO_{Bio}) + DIFF(DO_{Bio})]$.*
  *We agree that the word 'ambient' could be misleading. Therefore, as suggested, we will correct the word 'ambient' to 'adjacent' in the revised manuscript.*

**Comment:** P1, line 25: As you haven't yet defined "modulation of physical processes" this phrase appears to be jargon and is unclear to the reader.

*Response:*

*We will provide a clear definition of the 'physical modulation method' and its associated terms in the revised*
*manuscript.*

*With respect to the phrase 'modulation of physical processes', it should be the modulation of biogeochemical*
*effects on the oxygen due to physical transport. In our study, we considered that DO concentrations could be affected*
*by the biogeochemical processes by two ways. Firstly, the biogeochemical processes can produce or consume oxygen*
*to change the local DO concentrations directly. Secondly, the biogeochemical processes can also change DO*
*concentrations in the adjacent grid cells and hence change the DO fluxes from these adjacent grid cell; as a result, the*
*DO concentrations will be affected by the biogeochemical processes occurred in the adjacent grid cells. We defined the*
*latter mechanism as the modulation of biogeochemical effects on the oxygen due to physical transport.*

*To make it clear, we will change "modulation of physical processes" to "modulation of biogeochemical effects on*
*the oxygen due to physical transport" accordingly in the revised manuscript.*

**Comment:** P1, line 26: The values presented here and following seem like they should be process rates like 4.31 mg
l-1 d-1. Or if this value is an average over some seasonal or annual period, please state this.

*Response:*

*This value (4.31 mg $L^{-1}$) represents the decrease in DO concentrations due to the sediment oxygen demand using*
*the physical modulation method. It is averaged over July-August 2006. Specifically, the value of 4.31 mg $L^{-1}$ indicates*
*that the two-month averaged bottom DO concentrations will be higher by 4.31 mg $L^{-1}$ if the effect of the sediment*
*oxygen demand on DO is ignored. As suggested, we will clarify that this value is one averaged over the two months in*
*the revised manuscript.*

**Comment:** P1, line 32-33: This sentence doesn't make sense. On the one hand it is stated that photosynthesis
increased the O2 concentration, but then it also increased the hypoxic area (decreased O2 concentration). Please
clarify.

*Response:*

*Yes, there is a mistake here. What it should be is that the hypoxic area is increased when turning off the*
*photosynthesis and water column respiration. This will be corrected in the revised manuscript.*

**Comment:** P2, lines 2-7: These sentences are not clear and appear unnecessary. I suggest deleting them and starting
the Introduction with line 8,

*Response:*

*As suggested, we will delete these sentences in the revised manuscript.*

**Comment:** P3 line4: "Pearl River Estuary", estuary should not be capitalized

*Response:*

*As suggested, we will change it to 'Pearl River estuary' in the revised manuscript.*

**Comment:** P3, line 18: Describe here what you mean by "physical modulation of biogeochemical terms".

*Response:*

*As suggested, we will provide a further description to the physical modulation method and its associated terms*

*here in the revised manuscript (please see response above).*

**Comment:** P3-8: Nice description of the modeling

*Response:*

*Thank the referee for the positive comments.*

**Comment:** P8, line 7: In the 'Model Validation' section you might also suggest some additional observations that could be measured to assess the modulation model. Perhaps oxygen isotopes or additional rate measures would be useful to validate the biogeochemical O2 terms.

*Response:*

*As suggested, we will provide more results for our model validation (e.g., water elevations, temperature and*

*salinity profiles, chlorophyll-a, primary production, and particulate organic carbon) and also put forward some*

*suggestions for further observations (e.g., oxygen isotopes) that could be used to evaluate the modulation model*

*performance in the revised manuscript.*

*In addition, our model results show that the spatial distribution and duration of hypoxia in the Pearl River*

*estuary were mainly controlled by the re-aeration process and the sediment oxygen demand. In order to make our*

*conclusions more persuasive, we have compared our simulated results associated with these two important*

*biogeochemical processes with historical observations (Table 2). Such model-data comparison is of great importance*

*but not enough at the current stage due to data insufficiency. Therefore, in the revised manuscript we will also suggest*

*that the observations related to the important biogeochemical processes affecting DO should be strengthened in the*

*near future.*

**Comment:** P11, line 29: unclear what '905 1 t' is

*Response:*

*Here it means 9,051 tons. We will change it to "9,051 tons" in the revised manuscript.*

**Comment:** P12, lines 4-6: I disagree with this statement. The O2 gradient driven by photosynthesis would only be uniform or small if the light were at levels saturating to photosynthesis throughout the water-column

*Response:*

*Okay, this sentence should be better stated. In our study, the oxygen gradient driven by photosynthesis was represented by the gradient of $DO_{Phot}$. We calculated the vertical diffusive fluxes of $DO_{Phot}$ and $DO_{Rea}$ and found that the diffusive flux of $DO_{Phot}$ (-0.04 mg $L^{-1}$ $day^{-1}$) was much smaller than that of $DO_{Rea}$ (-0.48 mg $L^{-1}$ $day^{-1}$) (Fig. 8c). This result indicated that the gradient of $DO_{Phot}$ was much smaller than that of $DO_{Rea}$ since the diffusion coefficients were the same. From Fig. 8a it can be seen that at the surface layer of the PRE, the rates of re-aeration and photosynthesis were estimated 0.55 and 0.28 mg $L^{-1}$ $day^{-1}$, respectively; at the middle layer, these two rates were estimated 0 and 0.14 mg $L^{-1}$ $day^{-1}$, respectively. The difference of photosynthesis between the surface and middle layers was much smaller than the difference of re-aeration, which means that the oxygen gradient driven by photosynthesis was also much smaller.*

*We will modify this sentence and include the discussion above in the revised manuscript.*

**Comment:** P15, line 27: Correct 'Fig.9921a and Fig.9921b

*Response:*

*Okay, we will correct this to "Fig.12" in the revised manuscript.*

**Comment:** P16, line 11: '8 km3',use units consistent with what you present from the Gulf of Mexico and PRE (km2)

*Response:*

*The Chesapeake Bay and the northern Gulf of Mexico are the two well-known and well-studied areas which suffer from sever water hypoxia. We believe that it could be interesting and helpful to compare the hypoxia and its controlling mechanism between the Pearl River estuary and these two areas. Generally, the hypoxic volume was frequently estimated and reported for the Chesapeake Bay, while the hypoxic area was frequently estimated and reported for the northern Gulf of Mexico. To facilitate the comparison, we will calculate the hypoxic area (~237 $km^2$) and hypoxic volume (0.1255 $km^3$) for the Pearl River estuary and compare them with the northern Gulf of Mexico and the Chesapeake Bay accordingly in the revised manuscript.*

**Comment:** Table 2: In the heading do you mean 'WCR' instead of 'respiration by phytoplankton', which is hard to measure in practice and which is usually small in comparison to WCR.

*Response:*

*In fact, it is the respiration rate that was calculated by subtracting the nitrification oxygen consumption rate from the total oxygen consumption rate, neither WCR nor the respiration by phytoplankton (as the referee mentioned, it is difficult to measure). This respiration rate was reported by He et al. (2014) and was used to validate our model. We will correct this in the revised manuscript.*

**Comment:** Fig. 1b: Did you mean to show the cross-section (side-view) as noted in the figure caption? Here it shows the map view of the river network.

*Response:*

*We realize that it could be misleading using "cross-section" here, which in fact denotes the computational grids for the 1-D river network model. To clarify this, we will change it to "computational cross-sections" in the revised manuscript.*

**Comment:** Fig. 5: In the caption state the bias is between the two models, RCA and modulation

*Response:*

*Yes. We will explain the bias in the caption of Fig. 5 in the revised manuscript.*

**Comment:** Fig 6: Perhaps I missed it in the text, but why do the two models differ? Please discuss in text.

*Response:*

*As suggested, we will discuss the reason for the differences between the two models in the revised manuscript. Theoretically, there should be no differences between the two models since the transport equation is a linear equation. However, in the numerical model, the partial differential equations are approximately represented by difference equations. Some nonlinear numerical methods were applied to restrain the errors caused by discretizing the partial differential equations. We found that the differences between the two models are caused by these nonlinear numerical methods. We have conducted a test by turning off these nonlinear methods and found that the differences between the two models are removed but the model results become inaccurate. We would like to emphasize that the differences between the two models in our study are small and have little impact on the analysis of the hypoxia.*

**Comment:** Fig 8e: define ABio and LBio in the caption Technical corrections: There are too many for me to enumerate here. Please have an English editor assist with fixing plural noun/verb issues and other grammatical mistakes.

*Response:*

*As suggested, we will have an English editor edit our manuscript and correct grammatical mistakes throughout the text.*

*Reference*

He, B., Dai, M., Zhai, W., Guo, X., and Wang, L.: Hypoxia in the upper reaches of the Pearl River Estuary and its maintenance mechanisms: A synthesis based on multiple year observations during 2000–2008, Marine Chemistry, 167, 13-24, 2014.

---

## Author Comment (AC2) · 23 Feb 2017

We wish to thank the referee for the detailed and constructive comments which will be helpful to the revision of our manuscript. According to the comments, we plan to make revisions and improvements as below: 1) to include more results of validations and comparison of model skills between our model and other published studies; 2) to give clearer descriptions and explanations about 'physical modulation method' and its associated concepts; 3) to have an English editor edit our manuscript and correct grammatical mistakes throughout the text.

Detailed responses to all comments are given below. (Responses are shown in *Italic Font*)

**Comment:** This study by Wang et al. present a numerical investigation of the hypoxia condition in one of the largest river estuary along the Western Pacific - the Pearl River Delta.

According to the author, the Pearl River has a similar nutrient load as that of the Mississippi and the Pearl river runs through a highly populated region. The importance of this study is beyond question. And the coupling between 1D

river model and 3D coastal model as well as the author's efforts of improving the ECOM model is plausible. However,

I found this manuscript is not well written and the author failed to explain his/her points throughout the manuscript.

The so- called "physical modulation method" is merely to isolate lateral DO from in-situ biological source and sinks.

In addition, throughout the manuscript the definition of ambient biogeochemical process is very vague and sometimes contradictive. The available observation data does not support the high frequency zone (HFZ) proposed by the author. I

do agree with the other reviewer that this manuscript needs a throughout edits of scientific writing. I have a lot of problems to understand what the author wants to convey.

*Response:*

*We would like to thank the referee for recognizing the importance of our study and the detailed comments. We*

*have carefully read these comments and summarized them into three main criticisms. We believe that most of these*

*criticisms were mainly due to the misunderstanding caused by improper use of words like 'ambient' (which should be*

*'adjacent') in the text and inadequate description on the 'physical modulation method'. We will provide a further*

*description to the 'physical modulation method' and its associated terms in the revised manuscript and improve our*

*manuscript according to these comments. Responses to the three main comments are given below.*

*Firstly, for the comments on the 'physical modulation method', it actually means the modulation of*

*biogeochemical effects on the oxygen due to physical transport. In our study, we applied this method to tease apart the*

*biogeochemical and physical aspects of DO transport by using numerical DO tracers and to further demonstrate the*

*spatial connection between processes occurring in different locations in the system. Specifically, we considered that*

*DO concentrations could be affected by the biogeochemical processes by two ways: 1) the biogeochemical processes*

*can produce or consume oxygen to change the local DO concentrations directly; 2) the biogeochemical processes can*

*also change DO concentrations in the adjacent grid cells and hence change the DO fluxes from these adjacent grid*

*cells; as a result, the DO concentrations will be affected by the biogeochemical processes occurred in the adjacent*

*grid cells. We defined the latter mechanism as the modulation of biogeochemical effects on the oxygen due to physical*

*transport.*

*Regarding the 'contributions of ambient biogeochemical processes', it means the advective and diffusive fluxes of*
*oxygen which is produced or consumed by the biogeochemical processes in the adjacent grid cells. We have realized*
*that the word 'ambient' is misleading and we will change it to 'adjacent' as suggested in the revised manuscript.*

*Secondly, as to the comments on the observation data, we would like to emphasize that our model reasonably*
*reproduced the spatial distributions of oxygen as can be seen from the comparisons between our model results and the*
*available observations. In addition, we compared our simulated results associated with two important biogeochemical*
*processes controlling hypoxia (i.e. the re-aeration process and the sediment oxygen demand) with historical*
*observations (Table 2) in order to make our conclusions more persuasive. We realize that the available data for*
*hypoxia is insufficient. In fact, we have collected the observation data from 1976 to 2009 (Fig. r1) and found that there*
*were very few observation data available near the Modaomen sub-estuary (where the high frequency zone is located)*
*on hand. However, hypoxia near the Modaomen sub-estuary has been reported in previous observational studies (Lin*
*et al., 2001;Cai et al., 2013) and modeling studies (Zhang and Li, 2010). The spatial extent and characteristics of*
*hypoxia in our study is consistent with these previous studies.*

*Nevertheless, we realize the necessity and importance to strengthen the relevant observations. Therefore, we will*
*put forward some suggestions for further observations near the Modaomen sub-estuary in the revised manuscript. We*
*believe that our study will provide a scientific basis and guidance for conducting these field observations. In addition,*
*we will also provide more results for our model validation (e.g., water elevations, temperature and salinity profiles,*
*chlorophyll-a, primary production, and particulate organic carbon, results can be seen from Fig. r2-r5 and Table r1)*
*in the revised manuscript.*

*Finally, for the comment on the language, we will have an English editor edit our manuscript and correct*
*grammatical mistakes throughout the text.*

**Comment:** First of all, the abstract contradicts with itself. The author first indicated DO is affected by both local
biological processes and DO fluxes from the ambient water (I would treat this as boundary condition). And the author
claims the later (ambient water) is unclear. Then in next paragraph, the author indicated that the re-aeration (air-sea, i.e.
surface) and sediment oxygen demand (bottom) are most important. Again, the author indicates the importance of the
re-aeration, followed by a statement saying "turning off the re-aeration leads to hypoxia…… to the west lower estuary".
However, the author immediately states that hypoxia was observed in west lower estuary. So my question is re-aeration
good or bad? Important or not important at all? Then, in the last part of the abstract, the author states photosynthesis is
less important. Yet right following this, the author indicates that in HFZ, photosynthesis cause an increase of hypoxia
area to 591 km2— it confusing me how photosynthesis would worsen hypoxia (should be respiration, right? But
respiration is not directly linked with photosynthesis).

*Response:*

*Firstly, as mentioned above, the contributions of adjacent biogeochemical processes mean the advective and*

*diffusive fluxes of oxygen which is produced or consumed by the biogeochemical processes in the adjacent grid cells, not the boundary conditions. We realize that the word 'ambient' is misleading and we will change it to 'adjacent' as suggested in the revised manuscript.*

*Secondly, regarding the re-aeration, it is an important source for the oxygen in the water column. In the manuscript (abstract, P1, line 29-30), we mentioned that turning off the re-aeration would lead to a significant expansion of hypoxic area with the occurrence of persistent hypoxia in the west lower estuary (where no hypoxia occurred in the realistic simulation). This means that the re-aeration is good for alleviating the hypoxia in the Pearl River estuary. Moreover, our oxygen budget analysis showed the re-aeration could affect the bottom layer through the vertical diffusion (Fig. 8c). As shown in the Fig. 8a, the re-aeration provided a large amount of oxygen to the surface water, leading to a strong vertical DO gradient between the surface and middle layers. As a result, the majority of oxygen entering into the surface layer by the re-aeration (represented by $DO_{Rea}$) was transported to the middle layer and ~28% could reach the bottom layer eventually (Fig. 8c).*

*Finally, regarding the photosynthesis, we realize that there is a mistake here. What it should be is that turning off the photosynthesis and water column respiration would cause an increase in the hypoxic area. We will correct this in the revised manuscript.*

**Comment:** Page 2, line 29-34. The author first indicate DO from ambient water body, followed by " Take the re-aeration as an example". How would these two processes be connected (one is lateral boundary and the other is upper boundary)? The author then mentioned, "given the mechanism remains unclear", which one mechanism? Why it is not clear? Also, I do not think DO flux from ambient water bodies belong to the biogeochemical processes only, it indeed a combination of both bio and physical. Page 3, line 14-15, the author compared PRE with Northern Gulf of Mexico. I think one of the core mechanisms here is the tidal range difference (M2 in PRE vs S1 in NGOM).

***Response:***

*Firstly, as mentioned above, the word 'ambient' is misleading and will be changed to 'adjacent' in the revised manuscript. Here we are referring to the DO fluxes from the adjacent water bodies affected by adjacent biogeochemical processes (not from lateral boundary). We took the re-aeration as an example to explain the contribution of adjacent biogeochemical processes on DO.*

*Secondly, regarding the statement "given the mechanism remains unclear", it is referring to the mechanism of how the adjacent biogeochemical processes affect DO concentrations. We claimed it unclear because traditional methods have not teased apart and quantified the contributions of biogeochemical and physical processes to DO concentrations. We will provide a further description on the mechanism we would like to convey in the revised manuscript.*

*Thirdly, we recognize that the contributions of adjacent biogeochemical process are indeed a combination of both biogeochemical and physical processes. As mentioned earlier, the 'contributions of adjacent biogeochemical processes' mean the advective and diffusive fluxes of oxygen which is produced or consumed by the biogeochemical processes in*

*the adjacent grid cells. We realize that this statement is misleading and we will change it to 'the contributions of adjacent biogeochemical processes due to physical transport' in the revised manuscript.*

*Finally, regarding the tidal range difference, we agree that it is an important factor affecting hypoxia. Currently, we are actually working on another study addressing the impact of the tides as well as other physical factors (e.g. river discharges, winds) on the hypoxia in the Pear River estuary.*

**Comment:** Page 3, line 17. I do not agree that the mechanism remains unclear, since just tow lines above the author indicated that previous study suggest sediment oxygen demand and stratification are two main reasons. The author should give a more persuasive justification for this study and I believe there are many indeed. ˇ

***Response:***

*Yes, we agree that the sediment oxygen demand and stratification are two important factors affecting the hypoxia in the Pearl River estuary. However, we note from our model results that the distributions of the sediment oxygen demand and hypoxia were inconsistent and there was a shift between the hypoxic zone and the high sediment oxygen demand areas. This inconsistency could also be found in previous studies (Zhang and Li, 2010) where however there was no further analysis or discussion on it. The shift between the hypoxic zone and the high sediment oxygen demand areas is caused by the modulation of biogeochemical effects on the oxygen due to physical transport. In our study, we introduced the 'physical modulation method' to investigate this process.*

**Comment:** Page 6. Line 3, what is RDOC, LDOC, ReDOC, ExDOC stands for?

***Response:***

*RDOC, LDOC, ReDOC, and ExDOC are the abbreviations for four state variables in the RCA water quality model. They represent refractory dissolved organic carbon (RDOC), labile dissolved organic carbon (LDOC), reactive dissolved organic carbon (ReDOC), and algal exudate dissolved organic carbon (ExDOC). The explanations for these abbreviations can be found in P6, line 3-4. To make it easier to notice, we will show these abbreviations using a table along with other main state variables and parameters for the RCA.*

**Comment:** Page 6. Line 15. The authors used a constant boundary condition to study the impact from ambient DO transport which worries me a lot!

***Response:***

*As mentioned earlier, we realize that the word 'ambient' could be misleading and will changed to 'adjacent'. What it means is the contributions of adjacent biogeochemical processes, not the contributions from the boundary conditions.*

*The river boundary conditions of water quality variables were specified using the monthly observations. The open boundary conditions were specified spatial constant according to limited observations.*

**Comment:** Page 7 Line 7. Should not re-aeration of DO a physical process instead of a biological one?

*Response:*

*Yes, the re-aeration should be a physical process and we will modify this statement in the revised manuscript.*

**Comment:** Page 7. I am confused why the author needs to give repeated information in equation (6)-(12). They simply divide DO in to DObc and DObio, and they can just use equation (13), that is enough.

*Response:*

*Okay, we agree that there is repeated information in equations (9)-(12) and we will delete these equations in the*

*revised manuscript. For equations (6)-(8) we argue that they should be kept in the text. Equations (6)-(7) are two*

*important transport equations for $DO_{BC}$ and $DO_{Bio}$, which explicitly demonstrate how to calculate these two variables.*

*Equation (8) is also needed to illustrate $DO_{Rea}$, $DO_{Phot}$, $DO_{WCR}$, and $DO_{SOD}$ as these variables were presented and*

*discussed in the manuscript.*

**Comment:** Page 8. Model validation, I think it is necessary to present more validation for both physics and water quality, specifically some time series comparison. The Taylor diagram could represent the correlation yet we are not sure how the model could resolve temporal variability.

*Response:*

*Firstly, as suggested, we will provide more results for our model validation (e.g., water elevations, temperature*

*and salinity profiles, chlorophyll-a,primary production, and particulate oxygen carbon) in the revised manuscript.*

*Regarding the time series comparison, there were no time series observations (except water elevations) available for*

*July-August 2006 (our study period). However, our datasets used for model validation include salinity and temperature*

*from 146 cruise sites and water qualities from 53 cruise sites. These data were collected at different times of July and*

*August 2006, with a time span of 39 days. The agreement between our simulations and observations shows that our*

*model can reasonably reproduce the temporal and spatial distribution of hydrodynamic and water quality variables in*

*the Pearl River estuary.*

**Comment:** Page 9. Section 3.2. The author presents the difference between RCA and the Physical Modulation method.

How about the difference between Physical modulation and the in-situ data? Is there any improvement compared with difference between RCA and in-situ data?

*Response:*

*Please note that the physical modulation method is not an improvement of RCA. It is a new method that we*

*applied to tease apart the biogeochemical and physical aspects of DO dynamics by using RCA and the numerical*

*oxygen tracers. Theoretically, there should be no differences between the two models since the transport equation is a*

*linear equation. However, in the numerical model, the partial differential equations are approximately represented by*

*difference equations. Some nonlinear numerical methods are introduced to restrain the errors caused by discretizing*

*the partial differential equations. We found that the differences between the two models are caused by these nonlinear numerical methods. We have conducted a test by turning off these nonlinear methods and found that the differences between the two models are removed but the model results become inaccurate. We would like to emphasize that the differences between the two models in our study are small and have little impact on the analysis of the hypoxia.*

**Comment:** Page 9. Line 30. "The agreement indicates *: : :*" which agreement?

*Response:*

*It is the agreement between the DO fluxes calculated by RCA and those calculated using the physical modulation method. We will clarify this in the revised manuscript.*

**Comment:** Page 10. Line 16. The HFZ refers to high frequency zone of areas with DO < 3 mg/L. But why the author states "indicating HFZ is most possible to form hypoxia"?

*Response:*

*We realize that this statement is repeated because by definition the HFZ (high frequency zone) refers to the area where hypoxia is most likely to occur. We will delete it in the revised version.*

**Comment:** Page 11. Line 6. What does the author mean by "it is encompassed by the isoline of 10%"?

*Response:*

*Here what it should be mean is that the HFZ is the area encompassed by the 10% isoline of hypoxic frequency. We will clarify this in the revised manuscript.*

**Comment:** Page 11. Line 7-16. Should not these abbreviation goes to the figure caption?

*Response:*

*Okay, these abbreviations will be removed in the figure caption in the revised manuscript.*

**Comment:** Page 13. Line 4. "Although*: : :*. However *: : :* "very awkward wording

*Response:*

*As suggested, we will modify this in the revised manuscript and have an English editor edit our manuscript.*

**Comment:** Page 13, Line 10. "Both of which contribute 79% and 25%.... "I do not understand what the author wants to convey here.

*Response:*

*We realize that there is a grammatical mistake here. The sentence should be corrected "the sediment oxygen demand and the re-aeration contribute 79% and 25% of vertical diffusive flux, respectively." We will correct it in the revised manuscript.*

**Comment:** Page 13. Line 19-20. From Figure 8e I could tell how horizontal and vertical advection balance each other.

*Response:*

*Please note that the results related to the horizontal and vertical advective fluxes of oxygen are shown in Fig. 8a,*

*not Fig. 8e. We will include a reference therein to Fig. 8a in the revised manuscript. Fig. 8a shows that the horizontal*

*advection brings oxygen to the bottom layer in the PRE while the vertical advection brings oxygen out of the bottom*

*layer. The horizontal advective flux of DO is close to the vertical advective flux of DO.*

**Comment:** Page 14. Line 10-11. The author mentioned "two distinct areas characterized by the high level of sediment oxygen demands". However I could not find it on figure 10a referred here (I only see one area delineated by the white line).

*Response:*

*Please note that Fig. 10a shows the spatial distribution of the sediment oxygen demand in the PRE. As shown in*

*Fig. 10a, the blue indicates high sediment oxygen demand and the red indicates low sediment oxygen demand. There*

*are two areas highlighted by the blue, one of which is located inside the Lingdingyang sub-estuary and the other is*

*located at the Modaomen sub-estuary. These two areas correspond to the 'two distinct areas characterized by the high*

*level of sediment oxygen demands'.*

*Regarding the area delineated by the white line, it represents the HFZ (please see the caption of Fig. 10a). In Fig.*

*10a the HFZ was overlapped with the sediment oxygen demand in order to explicitly compare the spatial distributions*

*between the sediment oxygen demand and the hypoxia. It can be seen that there is a shift between the hypoxic zone and*

*the high sediment oxygen demand areas as mentioned earlier, which is caused by the modulation of biogeochemical*

*effects on the oxygen due to physical transport.*

**Comment:**    Page 14. Line 21-22. I do not understand what does the author mean here "depth to water column respiration and photosynthesis". Or the author means depth of water column respiration/photosynthesis. Also why this estimation is underestimated?

*Response:*

*Firstly, regarding this sentence "we therefore add sediment oxygen demand which is divided by the depth to water*

*column respiration and photosynthesis to represent the gross depletion rate", it means that we used*

*"SOD/depth+WCR+Phot" to represent the gross depletion of oxygen. Here, the SOD (sediment oxygen demand) and*

*WCR (water column respiration) are negative to indicate that they are sinks for oxygen. In the revised manuscript, we*

*will clarify this sentence.*

*Secondly, in previous studies (Murrell and Lehrter, 2011;Shen et al., 2013) the sediment oxygen demand was*

*estimated using the sediment oxygen consumption divided by the depth below the pycnocline, assuming that the water*

*below the pycnocline is well-mixed. In the Pearl River estuary, the depth of pycnocline significantly varies in time and*

*space, which makes it difficult to determine. Alternatively, we estimated the sediment oxygen demand by using the*

*sediment oxygen consumption divided by the total depth of the water column. The result was underestimated compared to one calculated using the depth below the pycnocline. We will clarify this in the revised manuscript. It should be noted that this underestimated value in the Pearl River estuary is larger than in other hypoxic areas (e.g. the Chesapeake Bay and northern Gulf of Mexico) so that the underestimation has little impact on the qualitative comparisons between the Pearl River estuary and the other hypoxic areas.*

**Comment:** Page 14. Line 27-28. Why such decrease cannot be explained only by biogeochemical processes?

***Response:***

*Please note that in the text we mentioned "This indicates that only the biogeochemical processes are not sufficient to explain the hypoxia in the HFZ since the effects of physical processes are also important" (P14, Line 27-28).*

*According to our model results, the distributions of the sediment oxygen demand and hypoxia are inconsistent. There is a shift between the hypoxic zone and the high sediment oxygen demand areas as mentioned earlier, which is caused by the modulation of biogeochemical effects on the oxygen due to physical transport. We will include the discussion above in the manuscript to make it clear.*

**Comment:** Page 14. Line 34-36. I could not tell how "simulated DO distribution in the bottom is in reasonable agreement with DO distribution". If we look at the available DO observations in Fig.4. There are only 2 points in Figure 4(c) that shows low oxygen in the so-called HFZ zone.

***Response:***

*Please note that in the text we mentioned "Figure 10c shows the simulated $DO_{Bio}$ distribution in the bottom is in reasonable agreement with DO distribution ……"(P14, Line 34-36). Here, the distribution of $DO_{Bio}$ (Fig. 10c) was compared to hypoxia (Fig. 7 b, d) in order to demonstrate that DOBio could resemble the occurrence of hypoxia in the PRE.*

*For the comments on the observation data, please see responses to the three main criticisms above (we would like to emphasize that our model reasonably reproduced the spatial distributions of oxygen as can be seen from the comparisons between our model results and the available observations. In addition, we compared our simulated results associated with two important biogeochemical processes controlling hypoxia (i.e. the re-aeration process and the sediment oxygen demand) with historical observations (Table 2) in order to make our conclusions more persuasive. We realize that the available data for hypoxia is insufficient. In fact, we have collected the observation data from 1976 to 2009 (Fig. r1) and found that there were very few observation data available near the Modaomen sub-estuary (where the high frequency zone is located) on hand. However, hypoxia near the Modaomen sub-estuary has been reported in previous observational studies (Lin et al., 2001;Cai et al., 2013) and modeling studies (Zhang and Li, 2010). The spatial extent and characteristics of hypoxia in our study is consistent with these previous studies.*

**Comment:** Page 15. Line 2-3. I could not tell that the lowest DOsod is observed in the HFZ zone from Fig 4. Although the author mentioned Fig 11 here, but it is model result rather than in-situ observation.

*Response:*

*Please note that Fig. 11 shows the $DO_{SOD}$ concentrations averaged over July-August 2006 and Fig. 4 shows comparison of the simulated and observed DO concentrations during the cruise periods. These two figures show results for different time periods.*

**Comment:** Page 15. Line 25. Figure 9921a. I do not know what the author what to convey here.

*Response:*

*It should be "Fig. 12'. We will correct this to "Fig.12" in the revised manuscript.*

**Comment:** Page 16. Why "phosphorous limitation can not be convincing enough?" The author does not mention much about phosphorous budget before this.

*Response:*

*Yin et al. (2004) suggested that the occurrence of hypoxia in the Pearl River estuary was restricted due to phosphorous limitation because phosphorous limitation inhibited the complete utilization of nutrients and eventually limited oxygen consumption in the water column. In our study, we found that the sediment oxygen demand was the dominant source of oxygen consumption at the bottom layer in the Pearl River estuary, which was consistent with previous studies(Zhang and Li, 2010). The total oxygen consumption rate in the Pearl River estuary was higher than those in the Chesapeake Bay and the northern Gulf of Mexico while the hypoxia in the Pearl River estuary was less severe (see comparison in Table 3). This indicates that there should be a different mechanism rather than phosphorous limitation controlling the hypoxia in the Pearl River estuary.*

**Comment:** Page 16. Line 20-21. Again, I am not convinced that Fig.10c (model) shows the realistic condition (Fig. 4).

*Response:*

*Please note that Fig. 10c shows the $DO_{Bio}$ averaged over July-August 2006 and Fig. 4 shows comparison of the simulated and observed DO concentrations during the cruise periods. These two figures show results for different time periods.*

*In the text, the distribution of $DO_{Bio}$ (Fig. 10c) was compared to hypoxia (Fig. 7 b, d) in order to demonstrate that DOBio could resemble the occurrence of hypoxia in the PRE.*

**Comment:** Page 16. Line 28. "is comparable more important". Awkward wording.

*Response:*

*As suggested, we will modify this in the revised manuscript.*

**Comment:** Page 17. Line 17. The author used a constant boundary condition, how can he/she justify that "DO originating from the boundaries to the bottom of the PRE and HFZ"?

*Response:*

In our study, the physical modulation method was applied to tease apart the contributions of boundary conditions, physical and biogeochemical processes to the oxygen. The oxygen originating from the boundaries was represented by $DO_{BC}$. Fig. 8d and Fig. 9d showed that at the bottom of the PRE and HFZ, the horizontal advection fluxes of $DO_{BC}$ were ~0.07 and 0.90 mg $L^{-1}$ $day^{-1}$, respectively.

*Reference*

Cai, S. Q., Zheng, S., and Wei, X.: Progress on the hydrodynamic characteristics and the hypoxia phenomenon in the Pearl River Estuary, JOURNAL OF TROPICAL OCEANOGRAPHY (in Chinese with English abstract), 32, 1-8, 2013.

Guo, W., Ye, F., Xu, S., and Jia, G.: Seasonal variation in sources and processing of particulate organic carbon in the Pearl River estuary, South China, Estuarine, Coastal and Shelf Science, 167, Part B, 540-548, 2015.

Lin, H. Y., Liu, S., and Han, W. Y.: Potential Trigger CTB, from Seasonal Bottom Water Hypoxia in the Pearl River Estuary, Journal of Zhanjiang Ocean University (in Chinese with English abstract), 21, 25-29, 2001.

Murrell, M. C., and Lehrter, J. C.: Sediment and Lower Water Column Oxygen Consumption in the Seasonally Hypoxic Region of the Louisiana Continental Shelf, Estuaries & Coasts, 34, 912-924, 2011.

Shen, J., Hong, B., and Kuo, A. Y.: Using timescales to interpret dissolved oxygen distributions in the bottom waters of Chesapeake Bay, Limnology and Oceanography, 58, 2237-2248, 2013.

Ye, H., Chen, C., Sun, Z., Tang, S., Song, X., Yang, C., Tian, L., and Liu, F.: Estimation of the Primary Productivity in Pearl River Estuary Using MODIS Data, Estuaries and Coasts, 38, 506-518, 2015.

Yin, K. D., Lin, Z. F., and Ke, Z. Y.: Temporal and spatial distribution of dissolved oxygen in the Pearl River Estuary and adjacent coastal waters, Continental Shelf Research, 24, 1935-1948, 2004.

Zhang, H., and Li, S. Y.: Effects of physical and biochemical processes on the dissolved oxygen budget for the Pearl River Estuary during summer, Journal of Marine Systems, 79, 65-88, 2010.

*Table r1 Comparisons between the simulated and observed chlorophyll-a, primary production, and particular organic carbon (POC)*

|  | Simulated | Historical observed | period |
|---|---|---|---|
| Chl-a (µg $L^{-1}$) | 1.92±1.96 | 1.64[a] | June 2012 |
| PP (mg $m^2$ $day^{-1}$) | 310.8±427.5 | 302.9[a] | June 2012 |
| POC (mg $L^{-1}$) | <0.5-3.01 | 0.40~>2.50[b] | November 2013, February 2014 May 2014 and August 2014 |

[a] *Ye et al. (2015)*

[b] *Guo et al. (2015)*

[Figure]

*Fig. r1 Survey stations of oxygen in the Pearl River estuary from 1976 to 2009*

[Figure]

[Figure]

*Fig. r2 Model-data comparisons of water elevation for 8 tidal stations (July 2006; the locations for these stations will*
*be shown in Fig. 3a in the revised manuscript)*

[Figure]

*Fig. r3 Model-data comparisons of salinity at the surface and bottom layers (July and August 2006)*

[Figure]

*Fig. r4 Model-data comparison of temperature at the surface and bottom layers (July and August 2006)*

[Figure]

*Fig. r5 Model-data comparisons of salinity (blue) and temperature (red) profiles at 48 stations (July and August 2006;*
*the locations for these stations are shown in Fig. r6)*

[Figure]

*Fig. r6 Survey stations of water elevation (red triangles), salinity and temperature (black circles), salinity and temperature profiles shown in Fig. r5 (black solid circles), and water quality variables (red crosses) during July and August 2006*

---

## Author Response (AR1)

We wish to thank the referee for the detailed and constructive comments which are very helpful to the revision of our manuscript. According to the comments, we have made revisions and improvements as below: 1) to include more results of validations and comparison of model skills between our model and other published studies; 2) to give clearer descriptions and explanations about 'physical modulation method' and its associated concepts; 3) to have an English editor edit our manuscript and correct grammatical mistakes throughout the text. The editing certificate and a marked-up manuscript version were provided at the end of this file.

Detailed responses to all comments are given below. (Responses are shown in *Italic Font*)

**Referee #1**

**Comment:** This paper reports on the use of a three-dimensional modeling system to explore the biogeochemical and physical mechanisms regulating O2 dynamics and bottom water hypoxia in the Pearl River estuary. A novelty of the study is the use of numerical O2 tracers to quantify the source and sink processes dictating O2 concentrations. The use of these tracers benefitted a diagnostic O2 mass balance in this shallow and dynamic estuary. Further, the study demonstrates the spatial connection between processes occurring in different locations in the system. From this analysis, the conclusion is that air-sea mixing and sediment oxygen demand were the primary processes regulating bottom-water O2 concentration and hypoxia. This is a different result than other well studied systems where hypoxia occurs such as Chesapeake Bay and northern Gulf of Mexico. Overall, I think this is an interesting paper that for the first time teases apart the biogeochemical and physical aspects of O2 dynamics in a shallow, river-dominated system.

Two major issues for the paper are the grammar and imprecise/incorrect use of language. I had to read many sentences two or three times for the meaning to shine through. These issues should be resolved by having a fluent English speaker edit the draft.

*Response:*

*Thank the referee for the positive comments. As suggested, we have had an English editor edit the manuscript.*

**Comment:** The use of 'ambient' here and throughout the paper is confusing to me. Do you mean to use 'adjacent'? It seems like you are inferring the advective and diffusive fluxes from adjacent grid cells. Please clarify.

*Response:*

*Yes, the contributions of ambient biogeochemical processes (was changed into 'the contributions of adjacent*

*source and sink processes (CAS)') represent the advective and diffusive fluxes of DO which is produced or consumed*

*by the source and sink processes in the adjacent grid cells. According to Eq. 9, the contributions of ambient source and*

*sink processes are described as* $\Delta t \times [-ADV(DO_S) + DIFF(DO_S)]$.

*We agree that the word 'ambient' could be misleading. Therefore, as suggested, we corrected the word 'ambient'*

*to 'adjacent' throughout the revised manuscript. We also clarified this phrase in the revised manuscript (Page 8, line*

*4-7).*

**Comment:** P1, line 25: As you haven't yet defined "modulation of physical processes" this phrase appears to be jargon and is unclear to the reader.

***Response:***

*We provided a clear definition of the 'physical modulation method' and its associated terms in Section 2.3 in the*
*revised manuscript.*

*With respect to the phrase 'modulation of physical processes', it should be the modulation of the effects of source*
*and sink processes on DO due to physical transport. In our study, we considered that DO concentrations could be*
*affected by the source and sink processes in two ways. Firstly, the source and sink processes can produce or consume*
*oxygen to change the local DO concentrations directly. Secondly, the source and sink processes can also change DO*
*concentrations in the adjacent grid cells and hence change the DO fluxes from these adjacent grid cell; as a result, the*
*DO concentrations will be affected by the source and sink processes occurred in the adjacent grid cells. We defined the*
*latter mechanism as the modulation of the effects of source and sink processes on the DO due to physical transport.*

*To make it clear, we changed "modulation of physical processes" to "modulation of the effects of source and sink*
*processes on DO due to physical transport" accordingly throughout the revised manuscript.*

**Comment:** P1, line 26: The values presented here and following seem like they should be process rates like 4.31 mg
l-1 d-1. Or if this value is an average over some seasonal or annual period, please state this.

***Response:***

*This value (4.31 mg $L^{-1}$) represents the decrease in DO concentrations due to the sediment oxygen demand using*
*the physical modulation method. It is averaged over July-August 2006. Specifically, the value of 4.31 mg $L^{-1}$ indicates*
*that the two-month averaged bottom DO concentrations will be higher by 4.31 mg $L^{-1}$ if the effect of the sediment*
*oxygen demand on DO is ignored. As suggested, we have clarified this in the revised manuscript (Page 1, line 16-17).*

**Comment:** P1, line 32-33: This sentence doesn't make sense. On the one hand it is stated that photosynthesis
increased the O2 concentration, but then it also increased the hypoxic area (decreased O2 concentration). Please
clarify.

***Response:***

*Yes, there is a mistake here. What it should be is that the hypoxic area is increased when turning off the*
*photosynthesis and water column respiration. This has been corrected in the revised manuscript (Page 1, line 27-29).*

**Comment:** P2, lines 2-7: These sentences are not clear and appear unnecessary. I suggest deleting them and starting
the Introduction with line 8,

***Response:***

*As suggested, we deleted these sentences in the revised manuscript.*

**Comment:** P3 line4: "Pearl River Estuary", estuary should not be capitalized

***Response:***

*As suggested, we have changed it to 'Pearl River estuary' throughout the revised manuscript.*

**Comment:** P3, line 18: Describe here what you mean by "physical modulation of biogeochemical terms".

*Response:*

*As suggested, we provided a description to the physical modulation method here in the revised manuscript (Page 3, line 10-17).*

**Comment:** P3-8: Nice description of the modeling

*Response:*

*Thank the referee for the positive comments.*

**Comment:** P8, line 7: In the 'Model Validation' section you might also suggest some additional observations that could be measured to assess the modulation model. Perhaps oxygen isotopes or additional rate measures would be useful to validate the biogeochemical O2 terms.

*Response:*

*As suggested, we provided more results for our model validation (e.g., water elevations, temperature and salinity profiles, chlorophyll-a, primary production, and particulate organic carbon; Page 8, line 28-29; Page 9, line 1-7; Page 9, line 28-30;) and also put forward some suggestions for further observations (e.g., oxygen isotopes, artificial tracers) that could be used to evaluate the physical modulation model performance in the revised manuscript (Page 10, line 29-31).*

*In addition, our model results show that the spatial distribution and duration of hypoxia in the Pearl River estuary were mainly controlled by the re-aeration process and the sediment oxygen demand. In order to make our conclusions more persuasive, we have compared our simulated results associated with these two important biogeochemical processes with historical observations (new Table 2). Such model-data comparison is of great importance but not enough at the current stage due to data insufficiency. Therefore, in the revised manuscript we will also suggest that the observations related to the important biogeochemical processes affecting DO should be strengthened in the near future (Page 9, line 31-32; Page 10, line 1-4).*

**Comment:** P11, line 29: unclear what '905 l t' is

*Response:*

*Here it means 9,051 tonnes. We have changed it to "9,051 tonnes" in the revised manuscript (Page 12, line 19).*

**Comment:** P12, lines 4-6: I disagree with this statement. The O2 gradient driven by photosynthesis would only be uniform or small if the light were at levels saturating to photosynthesis throughout the water-column

*Response:*

*Okay, this sentence should be better stated. In our study, the DO gradient driven by photosynthesis was represented by the gradient of $DO_{Phot}$. We calculated the vertical diffusive fluxes of $DO_{Phot}$ and $DO_{Rea}$ and found that the diffusive flux of $DO_{Phot}$ (-0.04 mg $L^{-1}$ $day^{-1}$) was much smaller than that of $DO_{Rea}$ (-0.48 mg $L^{-1}$ $day^{-1}$) (new Fig. 11c).*

*This result indicated that the gradient of $DO_{Phot}$ was much smaller than that of $DO_{Rea}$ since the diffusion coefficients*

*were the same. From new Fig. 11a it can be seen that at the surface layer of the PRE, the rates of re-aeration and*

*photosynthesis were estimated 0.55 and 0.28 mg $L^{-1}$ $day^{-1}$, respectively; at the middle layer, these two rates were*

*estimated 0 and 0.14 mg $L^{-1}$ $day^{-1}$, respectively. The difference of photosynthesis between the surface and middle layers*

*was much smaller than the difference of re-aeration, which means that the oxygen gradient driven by photosynthesis*

*was also much smaller.*

*We have modified this sentence and include the discussion above in the revised manuscript (Page 12, line 24-27;*

*Page 13, line 1).*

**Comment:** P15, line 27: Correct 'Fig.9921a and Fig.9921b

***Response:***

*Okay, we corrected this to "Fig.15" in the revised manuscript .*

**Comment:** P16, line 11: '8 km3',use units consistent with what you present from the Gulf of Mexico and PRE (km2)

***Response:***

*The Chesapeake Bay and the northern Gulf of Mexico are the two well-known and well-studied areas which suffer*

*from sever water hypoxia. We believe that it could be interesting and helpful to compare the hypoxia and its controlling*

*mechanism between the Pearl River estuary and these two areas. Generally, the hypoxic volume was frequently*

*estimated and reported for the Chesapeake Bay, while the hypoxic area was frequently estimated and reported for the*

*northern Gulf of Mexico. To facilitate the comparison, we calculated the hypoxic area (~237 $km^2$) and hypoxic volume*

*(0.1255 $km^3$) for the Pearl River estuary and compared them with the northern Gulf of Mexico and the Chesapeake*

*Bay accordingly in the revised manuscript (New equation (12); Page 12, line 3-5; Page 16, line 18-22).*

**Comment:** Table 2: In the heading do you mean 'WCR' instead of 'respiration by phytoplankton', which is hard to measure in practice and which is usually small in comparison to WCR.

***Response:***

*In fact, it is the respiration rate that was calculated by subtracting the nitrification oxygen consumption rate from*

*the WCR, neither WCR nor the respiration by phytoplankton (as the referee mentioned, it is difficult to measure). This*

*respiration rate was reported by He et al. (2014) and was used to validate our model. We have clarified and corrected*

*this in the revised manuscript (Page 9, line 24; Page 23, line 2).*

**Comment:** Fig. 1b: Did you mean to show the cross-section (side-view) as noted in the figure caption? Here it shows the map view of the river network.

***Response:***

*We realize that it could be misleading using "cross-section" here, which in fact denotes the computational grids*

*for the 1-D river network model. To clarify this, we changed it to "computational cross-sections" in the revised*

*manuscript (Page 24, line 3; new Fig. 1b).*

**Comment:** Fig. 5: In the caption state the bias is between the two models, RCA and modulation

*Response:*

*As suggested, we explained the bias in the caption of Fig. 5 (new Fig. 8) in the revised manuscript (Page 30, line*
*2-3).*

**Comment:** Fig 6: Perhaps I missed it in the text, but why do the two models differ? Please discuss in text.

*Response:*

*As suggested, we discussed the reason for the differences between the two models in the revised manuscript*
*(Page 10, line 6-7; Page 10, line 13-16). Theoretically, there should be no differences between the two models since*
*the transport equation is a linear equation. However, in the numerical model, the partial differential equations are*
*approximately represented by difference equations. Some nonlinear numerical methods were applied to restrain the*
*errors caused by discretizing the partial differential equations. We found that the differences between the two models*
*are caused by these nonlinear numerical methods. We have conducted a test by turning off these nonlinear methods*
*and found that the differences between the two models are removed but the model results become inaccurate. We would*
*like to emphasize that the differences between the two models in our study are small and have little impact on the*
*analysis of the hypoxia.*

**Comment:** Fig 8e: define ABio and LBio in the caption Technical corrections: There are too many for me to
enumerate here. Please have an English editor assist with fixing plural noun/verb issues and other grammatical
mistakes.

*Response:*

*As suggested, we had an English editor edit our manuscript and correct grammatical mistakes throughout the text. The*
*editing certificate was provided at the end of this file.*

*Reference*

He, B., Dai, M., Zhai, W., Guo, X., and Wang, L.: Hypoxia in the upper reaches of the Pearl River Estuary and its
maintenance mechanisms: A synthesis based on multiple year observations during 2000–2008, Mar. Chem., 167,
13-24, 2014.

**Referee #2**

**Comment:** This study by Wang et al. present a numerical investigation of the hypoxia condition in one of the largest
river estuary along the Western Pacific - the Pearl River Delta.

According to the author, the Pearl River has a similar nutrient load as that of the Mississippi and the Pearl river runs
through a highly populated region. The importance of this study is beyond question. And the coupling between 1D
river model and 3D coastal model as well as the author's efforts of improving the ECOM model is plausible. However,
I found this manuscript is not well written and the author failed to explain his/her points throughout the manuscript.

The so- called "physical modulation method" is merely to isolate lateral DO from in-situ biological source and sinks. In addition, throughout the manuscript the definition of ambient biogeochemical process is very vague and sometimes contradictive. The available observation data does not support the high frequency zone (HFZ) proposed by the author. I do agree with the other reviewer that this manuscript needs a throughout edits of scientific writing. I have a lot of problems to understand what the author wants to convey.

***Response:***

*We would like to thank the referee for recognizing the importance of our study and the detailed comments. We have carefully read these comments and summarized them into three main criticisms. We believe that most of these criticisms were mainly due to the misunderstanding caused by improper use of words like 'ambient' (which should be 'adjacent') in the text and inadequate description on the 'physical modulation method'. We provided a further description to the 'physical modulation method' and its associated terms in the revised manuscript and improved our manuscript according to these comments. Responses to the three main comments are given below.*

*Firstly, for the comments on the 'physical modulation method', it actually means the modulation of the effects of source and sink processes on DO due to physical transport. In our study, we applied this method to tease apart the contributions of boundary conditions as well as the source and sink processes to DO transport flux by using numerical DO tracers and to further demonstrate the spatial connection between the source and sink processes occurring in different locations in the system. Specifically, we considered that DO concentrations could be affected by the source and sink processes in two ways: 1) the source and sink processes can produce or consume oxygen to change the local DO concentrations directly; 2) the source and sink processes can also change DO concentrations in the adjacent grid cells and hence change the DO fluxes from these adjacent grid cells; as a result, the DO concentrations will be affected by the source and sink processes occurred in the adjacent grid cells. We defined the latter mechanism as the modulation of the effects of the source and sink processes on the DO due to physical transport (Page 3, line 10-17; Page 7, line 3-4; Page 8, line 2-10, new equation (9)).*

*Regarding the 'contributions of ambient biogeochemical processes' (has been changed into 'contributions of adjacent source and sink processes' in the revised manuscript), it means the advective and diffusive fluxes of oxygen which is produced or consumed by the source and sink processes in the adjacent grid cells. We have realized that the word 'ambient' is misleading and we changed it to 'adjacent' as suggested throughout the revised manuscript. We also clarified the 'contributions of adjacent source and sink processes' in the revised manuscript (Page 8, line 4-7).*

*Secondly, as to the comments on the observation data, we would like to emphasize that our model reasonably reproduced the spatial distributions of physical variables (including water levels, salinity, and temperature) and DO as can be seen from the comparisons between our model results and the available observations (new Fig. 3-7). In addition, we compared our simulated results associated with two important biogeochemical processes controlling hypoxia (i.e. the re-aeration process and the sediment oxygen demand) with historical observations (new Table 3) in order to make our conclusions more persuasive. We realize that the available data for hypoxia is insufficient. In fact, we have collected the observation data from 1976 to 2009 (Fig. r1) and found that there were very few observation data available near the Modaomen sub-estuary (where the high frequency zone is located) on hand. However, hypoxia near the Modaomen sub-estuary has been reported in previous observational studies (Lin et al., 2001;Cai et al., 2013)*

*and modeling studies (Zhang and Li, 2010). The spatial extent and characteristics of hypoxia in our study is consistent*
*with these previous studies.*
*Nevertheless, we realize the necessity and importance to strengthen the relevant observations. Therefore, we put*
*forward some suggestions for further observations near the Modaomen sub-estuary in the revised manuscript (Page 9,*
*line 31-32; Page 10, line 1-4). We believe that our study will provide a scientific basis and guidance for conducting*
*these field observations. In addition, we also provided more results for our model validation (e.g., water elevations,*
*temperature and salinity profiles, chlorophyll-a, primary production, and particulate organic carbon, results can be*
*seen from new Fig. 3-7 and new Table 3) in the revised manuscript ( Page 8, line 28-29; Page 9, line 1-7; Page 9, line*
*28-30).*
*Finally, for the comment on the language, we had an English editor edit our manuscript and correct grammatical*
*mistakes throughout the text. We also provided the editing certificate at the end of this file.*
**Comment:** First of all, the abstract contradicts with itself. The author first indicated DO is affected by both local
biological processes and DO fluxes from the ambient water (I would treat this as boundary condition). And the author
claims the later (ambient water) is unclear. Then in next paragraph, the author indicated that the re-aeration (air-sea, i.e.
surface) and sediment oxygen demand (bottom) are most important. Again, the author indicates the importance of the
re-aeration, followed by a statement saying "turning off the re-aeration leads to hypoxia…… to the west lower estuary".
However, the author immediately states that hypoxia was observed in west lower estuary. So my question is re-aeration
good or bad? Important or not important at all? Then, in the last part of the abstract, the author states photosynthesis is
less important. Yet right following this, the author indicates that in HFZ, photosynthesis cause an increase of hypoxia
area to 591 km2— it confusing me how photosynthesis would worsen hypoxia (should be respiration, right? But
respiration is not directly linked with photosynthesis).
***Response:***
*Firstly, as mentioned above, the contributions of adjacent source and sink processes mean the advective and*
*diffusive fluxes of DO which is produced or consumed by the source and sink processes in the adjacent grid cells, not*
*the boundary conditions. We realize that the word 'ambient' is misleading and we have changed it to 'adjacent' as*
*suggested throughout the revised manuscript.*
*Secondly, regarding the re-aeration, it is an important source for DO in the water column. In P1, line 28-30 of the*
*original manuscript (P1, line 25-27 in revised manuscript), we mentioned that turning off the re-aeration would lead to*
*a significant expansion of hypoxic area with the occurrence of persistent hypoxia in the west lower estuary (where no*
*hypoxia occurred in the realistic simulation). This means that the re-aeration is good for alleviating the hypoxia in the*
*Pearl River estuary. Moreover, our DO budget analysis showed the re-aeration could affect the bottom layer through*
*the vertical diffusion (new Fig. 11c). As shown in the new Fig. 11a, the re-aeration provided a large amount of oxygen*
*to the surface water, leading to a strong vertical DO gradient between the surface and middle layers. As a result, the*
*majority of oxygen entering into the surface layer by the re-aeration (represented by $DO_{Rea}$) was transported to the*
*middle layer and ~28% could reach the bottom layer eventually (new Fig. 11c).*
*Finally, regarding the photosynthesis, we realize that there is a mistake here. What it should be is that turning off*

*the photosynthesis and water column respiration would cause an increase in the hypoxic area. We corrected this in the revised manuscript (Page 1, line 27-29).*

**Comment:** Page 2, line 29-34. The author first indicate DO from ambient water body, followed by " Take the re-aeration as an example". How would these two processes be connected (one is lateral boundary and the other is upper boundary)? The author then mentioned, "given the mechanism remains unclear", which one mechanism? Why it is not clear? Also, I do not think DO flux from ambient water bodies belong to the biogeochemical processes only, it indeed a combination of both bio and physical. Page 3, line 14-15, the author compared PRE with Northern Gulf of Mexico. I think one of the core mechanisms here is the tidal range difference (M2 in PRE vs S1 in NGOM).

*Response:*

*Firstly, as mentioned above, the word 'ambient' is misleading and has been changed to 'adjacent' in the revised manuscript. Here we were referring to the DO fluxes from the adjacent water bodies affected by adjacent source and sink processes (not from lateral boundary). We took the re-aeration as an example to explain the contribution of adjacent source and sink processes on DO.*

*Secondly, regarding the statement "given the mechanism remains unclear", it is referring to the mechanism of how the adjacent source and sink processes affect DO concentrations. We claimed it unclear because traditional methods have not teased apart and quantified the contributions of boundary conditions as well as the source and sink processes to DO concentrations. In the revised manuscript, we modified the logic of the introduction to better state the purpose of our study.*

*Thirdly, we recognize that the contributions of adjacent source and sink process are indeed a combination of both biogeochemical and physical processes. As mentioned earlier, the 'contributions of adjacent source and sink processes mean the advective and diffusive fluxes of DO which is produced or consumed by the source and sink processes in the adjacent grid cells. We realize that this statement is misleading and we changed it to 'the contributions of adjacent source and sink processes due to physical transport' in the revised manuscript.*

*Finally, regarding the tidal range difference, we agree that it is an important factor affecting hypoxia. Currently, we are actually working on another study addressing the impact of the tides as well as other physical factors (e.g. river discharges, winds) on the hypoxia in the Pear River estuary.*

**Comment:** Page 3, line 17. I do not agree that the mechanism remains unclear, since just two lines above the author indicated that previous study suggest sediment oxygen demand and stratification are two main reasons. The author should give a more persuasive justification for this study and I believe there are many indeed. ˇ

*Response:*

*Yes, we agree that the sediment oxygen demand and stratification are two important factors affecting the hypoxia in the Pearl River estuary. However, we note from our model results that the distributions of the sediment oxygen demand and hypoxia were inconsistent and there was a shift between the hypoxic zone and the high sediment oxygen demand areas. This inconsistency could also be found in previous studies (Zhang and Li, 2010) where however there was no further analysis or discussion on it. The shift between the hypoxic zone and the high sediment oxygen demand*

*areas is caused by the modulation of the effects of source and sink processes on DO due to physical transport. In the*

*revised manuscript, we have clarified this misinterpretation (Page 3, line 10-17).*

**Comment:** Page 6. Line 3, what is RDOC, LDOC, ReDOC, ExDOC stands for?

*Response:*

*RDOC, LDOC, ReDOC, and ExDOC are the abbreviations for four state variables in the RCA water quality*

*model. They represent refractory dissolved organic carbon (RDOC), labile dissolved organic carbon (LDOC), reactive*

*dissolved organic carbon (ReDOC), and algal exudate dissolved organic carbon (ExDOC). The explanations for these*

*abbreviations can be found in one sentence after these abbreviations (P6, line 3-4 in original manuscript). In the*

*revised manuscript, to make it easier to notice, we defined their meanings before using these abbreviations and showed*

*these abbreviations using a table along with other main state variables for the RCA and the physical modulation*

*method (Page 6, line 8-10; new Table 1).*

**Comment:** Page 6. Line 15. The authors used a constant boundary condition to study the impact from ambient DO

transport which worries me a lot!

*Response:*

*As mentioned earlier, we realize that the word 'ambient' could be misleading and we changed it to 'adjacent'.*

*What it means is the contributions of adjacent source and sink processes, not the contributions from the boundary*

*conditions.*

*The river boundary conditions of water quality variables were specified using the monthly observations. The open*

*boundary conditions were specified spatial constant according to limited observations.*

**Comment:** Page 7 Line 7. Should not re-aeration of DO a physical process instead of a biological one?

*Response:*

*Yes, the re-aeration should be a physical process and we modified this statement. In the revised manuscript, we*

*changed 'biogeochemical source and sink processes into 'the source and sink processes' (Page 7, line 12-13). Other*

*associated terms were also modified correspondingly in the revised manuscript.*

**Comment:** Page 7. I am confused why the author needs to give repeated information in equation (6)-(12). They simply divide DO in to DObc and DObio, and they can just use equation (13), that is enough.

*Response:*

*Okay, we agree that there is repeated information in equations (9)-(12) and we deleted these equations in the*

*revised manuscript. For equations (6)-(8) we argue that they should be kept in the text. Equations (6)-(7) are two*

*important transport equations for $DO_{BC}$ and $DO_s$, which explicitly demonstrate how to calculate these two variables.*

*Equation (8) is also needed to illustrate $DO_{Rea}$, $DO_{Phot}$, $DO_{WCR}$, and $DO_{SOD}$ as these variables were presented and*

*discussed in the manuscript (section 2.3).*

**Comment:** Page 8. Model validation, I think it is necessary to present more validation for both physics and water quality, specifically some time series comparison. The Taylor diagram could represent the correlation yet we are not sure how the model could resolve temporal variability.

***Response:***

*Firstly, as suggested, we provided more results for our model validation (e.g., water elevations, temperature and*

*salinity profiles, chlorophyll-a,primary production, and particulate oxygen carbon) in the revised manuscript (Page 8,*

*line 28-29; Page 9, line 1-7; Page 9, line 28-30; new Fig 3-7; new Table 3). Regarding the time series comparison,*

*there were no time series observations (except water elevations) available for July-August 2006 (our study period).*

*However, our datasets used for model validation include salinity and temperature from 146 cruise sites and water*

*qualities from 53 cruise sites. These data were collected at different times of July and August 2006, with a time span of*

*39 days. The agreement between our simulations and observations shows that our model can reasonably reproduce the*

*temporal and spatial distribution of hydrodynamic and water quality variables in the Pearl River estuary.*

**Comment:** Page 9. Section 3.2. The author presents the difference between RCA and the Physical Modulation method.

How about the difference between Physical modulation and the in-situ data? Is there any improvement compared with difference between RCA and in-situ data?

***Response:***

*Please note that the physical modulation method is not an improvement of RCA. It is a new method that we*

*applied to tease apart the contributions of boundary conditions as well as the source and sink processes to DO*

*dynamics by using RCA and the numerical DO tracers. Theoretically, there should be no differences between the two*

*models since the transport equation is a linear equation. However, in the numerical model, the partial differential*

*equations are approximately represented by difference equations. Some nonlinear numerical methods are introduced to*

*restrain the errors caused by discretizing the partial differential equations. We found that the differences between the*

*two models are caused by these nonlinear numerical methods. We have conducted a test by turning off these nonlinear*

*methods and found that the differences between the two models are removed but the model results become inaccurate.*

*We would like to emphasize that the differences between the two models in our study are small and have little impact*

*on the analysis of the hypoxia. In the revised manuscript, we have included the discussion above to explain the*

*differences between these two models (Page 10, line 6-7; Page 10, line 13-16).*

**Comment:** Page 9. Line 30. "The agreement indicates *:::*" which agreement?

***Response:***

*It is the agreement between the DO fluxes calculated by RCA and those calculated using the physical modulation*

*method. We have clarified this in the revised manuscript (Page 10, line 27-29).*

**Comment:** Page 10. Line 16. The HFZ refers to high frequency zone of areas with DO < 3 mg/L. But why the author states "indicating HFZ is most possible to form hypoxia"?

***Response:***

*We realize that this statement is repeated because by definition the HFZ (high frequency zone) refers to the area*

*where hypoxia is most likely to occur. We have deleted it in the revised version.*

**Comment:** Page 11. Line 6. What does the author mean by "it is encompassed by the isoline of 10%"?

***Response:***

*Here what it should be mean is that the HFZ is the area encompassed by the 10% isoline of hypoxic frequency. We*

*have clarified this in the revised manuscript (Page 11, line 12-13)*

**Comment:** Page 11. Line 7-16. Should not these abbreviation goes to the figure caption?

***Response:***

*As suggested, these abbreviations and their explanations have been added in the figure caption in the revised*

*manuscript (Page 33, line 7-15)*

**Comment:** Page 13. Line 4. "Although*: : :.* However *: : :* "very awkward wording

***Response:***

*As suggested, we have modified this in the revised manuscript and had an English editor edit our manuscript.*

**Comment:** Page 13, Line 10. "Both of which contribute 79% and 25%.... "I do not understand what the author wants to convey here.

***Response:***

*We realize that there is a grammatical mistake here. The sentence means that the sediment oxygen demand and*

*the re-aeration contribute 79% and 25% of vertical diffusive flux, respectively. We corrected it in the revised*

*manuscript (Page 13, line 25-27)*

**Comment:** Page 13. Line 19-20. From Figure 8e I could tell how horizontal and vertical advection balance each other.

***Response:***

*Please note that the results related to the horizontal and vertical advective fluxes of DO are shown in Fig. 8a*

*(new Fig. 11a), not Fig. 8e. We have included a reference therein to new Fig. 11a in the revised manuscript (Page 13,*

*line 30-32). New Fig. 11a shows that the horizontal advection brings oxygen to the bottom layer in the PRE while the*

*vertical advection brings oxygen out of the bottom layer. The horizontal advective flux of DO is close to the vertical*

*advective flux of DO.*

**Comment:** Page 14. Line 10-11. The author mentioned "two distinct areas characterized by the high level of sediment oxygen demands". However I could not find it on figure 10a referred here (I only see one area delineated by the white line).

***Response:***

*Please note that Fig. 10a (new Fig. 13a) shows the spatial distribution of the sediment oxygen demand in the PRE.*

*As shown in new Fig. 13a, the blue indicates high sediment oxygen demand and the red indicates low sediment oxygen*

*demand. There are two areas highlighted by the blue, one of which is located inside the Lingdingyang bay and the*

*other is located at the Modaomen sub-estuary. These two areas correspond to the 'two distinct areas characterized by*

*the high level of sediment oxygen demands'.*

*Regarding the area delineated by the white line, it represents the HFZ (please see the caption of new Fig. 13a). In*

*new Fig. 13a the HFZ was overlapped with the sediment oxygen demand in order to explicitly compare the spatial*

*distributions between the sediment oxygen demand and the hypoxia. It can be seen that there is a shift between the*

*hypoxic zone and the high sediment oxygen demand areas as mentioned earlier, which is caused by the modulation of*

*the effects of the source and sink processes on the DO due to physical transport.*

**Comment:** Page 14. Line 21-22. I do not understand what does the author mean here "depth to water column respiration and photosynthesis". Or the author means depth of water column respiration/photosynthesis. Also why this estimation is underestimated?

***Response:***

*Firstly, regarding this sentence "we therefore add sediment oxygen demand which is divided by the depth to water*

*column respiration and photosynthesis to represent the gross depletion rate", it means that we used*

*"SOD/depth+WCR+Phot" to represent the gross depletion of oxygen. Here, the SOD (sediment oxygen demand) and*

*WCR (water column respiration) are negative to indicate that they are sinks for DO. In the revised manuscript, we*

*have changed this sentence to make it clear (Page 14, line 29-30).*

*Secondly, in previous studies (Murrell and Lehrter, 2011;Shen et al., 2013) the sediment oxygen consumption rate*

*(mg $L^{-1}$ $day^{-1}$) was estimated using the sediment oxygen demand (g $m^{-2}$ $day^{-1}$) divided by the depth below the*

*pycnocline, assuming that the water below the pycnocline is well-mixed. In the Pearl River estuary, the depth of*

*pycnocline significantly varies in time and space, which makes it difficult to determine. In additional, the shallow*

*depth and large vertical velocities make it possible for SOD to affect the surface layer significantly. Therefore, we*

*alternatively estimated the sediment oxygen consumption rate (mg $L^{-1}$ $day^{-1}$) by using the sediment oxygen demand (g*

*$m^{-2}$ $day^{-1}$) divided by the total depth of the water column. The result was underestimated compared to one calculated*

*using the depth below the pycnocline. It should be noted that this underestimated value in the Pearl River estuary is*

*larger than in other hypoxic areas (e.g. the Chesapeake Bay and northern Gulf of Mexico) so that the underestimation*

*has little impact on the qualitative comparisons between the Pearl River estuary and the other hypoxic areas.*

**Comment:** Page 14. Line 27-28. Why such decrease cannot be explained only by biogeochemical processes?

***Response:***

*Please note that in the original text we mentioned "This indicates that only the biogeochemical processes are not*

*sufficient to explain the hypoxia in the HFZ since the effects of physical processes are also important" (P14, Line*

*27-28 in original manuscript).*

*According to our model results, the distributions of the sediment oxygen demand and hypoxia are inconsistent.*

*There is a shift between the hypoxic zone and the high sediment oxygen demand areas as mentioned earlier, which is*

*caused by the modulation of the effects of the source and sink processes on DO due to physical transport. We rewrote*

*the discussion above in the revised manuscript for a better statement (Page 14, line 31; Page 15, line 1-3).*

**Comment:** Page 14. Line 34-36. I could not tell how "simulated DO distribution in the bottom is in reasonable agreement with DO distribution". If we look at the available DO observations in Fig.4. There are only 2 points in

Figure 4(c) that shows low oxygen in the so-called HFZ zone.

*Response:*

*Please note that in the original text we mentioned "Figure 10c shows the simulated $DO_{Bio}$ distribution in the*

*bottom is in reasonable agreement with DO distribution ……"(P14, Line 34-36 in original manuscript). Here, the*

*distribution of $DO_{Bio}$ (has been changed into $DO_S$ in the revised manuscript; new Fig. 13c) was compared to hypoxia*

*(new Fig. 10b, d) in order to demonstrate that $DO_S$ could resemble the occurrence of hypoxia in the PRE.*

*For the comments on the observation data, please see responses to the three main criticisms above (we would like*

*to emphasize that our model reasonably reproduced the spatial distributions of oxygen as can be seen from the*

*comparisons between our model results and the available observations. In addition, we compared our simulated*

*results associated with two important biogeochemical processes controlling hypoxia (i.e. the re-aeration process and*

*the sediment oxygen demand) with historical observations (new Table 3) in order to make our conclusions more*

*persuasive. We realize that the available data for hypoxia is insufficient. In fact, we have collected the observation data*

*from 1976 to 2009 (Fig. r1) and found that there were very few observation data available near the Modaomen*

*sub-estuary (where the high frequency zone is located) on hand. However, hypoxia near the Modaomen sub-estuary*

*has been reported in previous observational studies (Lin et al., 2001;Cai et al., 2013) and modeling studies (Zhang*

*and Li, 2010). The spatial extent and characteristics of hypoxia in our study is consistent with these previous studies.*

**Comment:** Page 15. Line 2-3. I could not tell that the lowest DOsod is observed in the HFZ zone from Fig 4. Although the author mentioned Fig 11 here, but it is model result rather than in-situ observation.

*Response:*

*Please note that Fig. 11 (new Fig. 14) shows the $DO_{SOD}$ concentrations averaged over July-August 2006 and Fig.*

*4 (new Fig. 7) shows comparison of the simulated and observed DO concentrations during the cruise periods. These*

*two figures show results for different time periods.*

**Comment:** Page 15. Line 25. Figure 9921a. I do not know what the author what to convey here.

*Response:*

*We have corrected this to "Fig.15" in the revised manuscript.*

**Comment:** Page 16. Why "phosphorous limitation can not be convincing enough?" The author does not mention much about phosphorous budget before this.

*Response:*

*Yin et al. (2004) suggested that the occurrence of hypoxia in the Pearl River estuary was restricted due to*

*phosphorous limitation because phosphorous limitation inhibited the complete utilization of nutrients and eventually*

*limited oxygen consumption in the water column. In our study, we found that the sediment oxygen demand was the*
*dominant source of oxygen consumption at the bottom layer in the Pearl River estuary, which was consistent with*
*previous studies(Zhang and Li, 2010). In addition, the total oxygen consumption rate in the Pearl River estuary was*
*higher than those in the Chesapeake Bay and the northern Gulf of Mexico while the hypoxia in the Pearl River estuary*
*was less severe (see comparison in new Table 4). This indicates that there should be a different mechanism rather than*
*phosphorous limitation controlling the hypoxia in the Pearl River estuary. In the revised manuscript, we have rewritten*
*the above discussion for a better statement (Page 16, line 11-23).*

**Comment:** Page 16. Line 20-21. Again, I am not convinced that Fig.10c (model) shows the realistic condition (Fig. 4).
*Response:*
*Please note that Fig. 10c (new Fig. 13) shows the $DO_{Bio}$ (has been changed into $DO_S$ in the revised manuscript)*
*averaged over July-August 2006 and Fig. 4 (new Fig. 7) shows comparison of the simulated and observed DO*
*concentrations during the cruise periods. These two figures show results for different time periods.*
*In the text, the distribution of DOs (new Fig. 13c) was compared to hypoxia (new Fig. 10b, d) in order to*
*demonstrate that DOs could resemble the occurrence of hypoxia in the PRE.*

**Comment:** Page 16. Line 28. "is comparable more important". Awkward wording.
*Response:*
*As suggested, we have modified this in the revised manuscript and we had an English editor edit our manuscript.*

**Comment:** Page 17. Line 17. The author used a constant boundary condition, how can he/she justify that "DO
originating from the boundaries to the bottom of the PRE and HFZ"?
*Response:*
*In our study, the physical modulation method was applied to tease apart the contributions of boundary conditions*
*as well as source and sink processes to the DO. The oxygen originating from the boundaries was represented by $DO_{BC}$.*
*Fig. 8d (new Fig. 11d) and Fig. 9d (new Fig. 12d) showed that at the bottom of the PRE and HFZ, the horizontal*
*advection fluxes of $DO_{BC}$ were ~0.07 and 0.90 mg $L^{-1}$ $day^{-1}$, respectively.*

**Editing certificate**

**CERTIFICATE OF**
**ENGLISH EDITING**

This document certifies that the paper listed below has been edited to ensure that the language is clear and free of errors. The logical presentation of ideas and the structure of the paper were also checked during the editing process. The edit was performed by professional editors at Editage, a division of Cactus Communications. The intent of the author's message was not altered in any way during the editing process. The quality of the edit has been guaranteed, with the assumption that our suggested changes have been accepted and have not been further altered without the knowledge of our editors.

**TITLE OF THE PAPER**
A numerical analysis of biogeochemical controls with physical modulation on hypoxia during summer in the Pearl River estuary

**AUTHORS**
Bin Wang, Jiatang Hu, Shiyu Li, Dehong Liu

**JOB CODE**
KKEVG_1_2

[Figure]

Signature

[Figure]

Vikas Narang,
Vice President, Author Services, Editage

Date of Issue
**April 04, 2017**

Editage, a brand of Cactus Communications, offers professional English language editing and publication support services to authors engaged in over 500 areas of research. Through its community of experienced editors, which includes doctors, engineers, published scientists, and researchers with peer review experience, Editage has successfully helped authors get published in internationally reputed journals. Authors who work with Editage are guaranteed excellent language quality and timely delivery.

CACTUS®

**Contact Editage**

[revised manuscript text omitted]

---

## Author Response (AR2)

Review of "A numerical analysis of biogeochemical controls with physical modulation on hypoxia during summer in the Pearl River Estuary" by Wang et al. (Revised).

The manuscript has been substantially revised and edited based on the previous review comments, and is now clearer and more easily read. The description of the physical modulation method, which was developed for this study, is much improved. Further, the authors have added new model validation results in comparison to field observations from the Pearl River estuary and added comparisons of modeled metabolism processes with values reported in the literature. These results support and strengthen the overall conclusions of the paper. I commend the authors for this thorough revision. Some specific comments follow.

*Response:*

*We wish to thank the referee for the positive comments on our manuscript and we will make revisions according to these comments. Detailed responses to all comments are given below. (Responses are shown in Italic Font).*

**Specific comments:**

P1, line 24: Standard convention is to omit the space before a % sign. Please correct throughout.

*Response: Okay, we have omitted the space before the % sign throughout the revised manuscript.*

P2, lines 21-23: For the sentence beginning "Shen et al. (2013) …" it is not clear to me how this sentence contributes to the paragraph. Please clarify.

*Response: Here we intended to use the study by Shen et al. (2013) as an example to demonstrate that elucidating the relative contributions of physical and biogeochemical processes on hypoxia is an important question. In the revised manuscript, we deleted this sentence to make this section clearer and more easily read.*

P4, line 15: Specify the Mellor-Yamada turbulent closure model that was used. I think it is MY 2.5 in ECOMSED

*Response: Yes, it should be the Mellor and Yamada's level 2.5 turbulent closure model. We have corrected it in the revised manuscript.*

P5, line 8: Insert "suspended" before "sediments" and insert "which" before "significantly"

*Response: We have revised the sentence as suggested.*

P15, line 18: Change "conductive" to "conducive"

*Response: As suggested, we have changed the word 'conductive' to 'conducive'*

P16, lines 24-25: This sentence is redundant of the first paragraph in section 4.4 and could be omitted.

*Response: Okay, we have deleted this sentence as suggested.*

P19, line 18: M.C. Murrell is included twice as an author in this reference

***Response:*** *Thanks! We have modified the reference in the revised manuscript.*

P23, line 15: Citation should be Murrell and Lehrter (2011)

***Response:*** *We modified this citation in the revised manuscript.*

P32, Figure 10 c,d: Frequency is misspelled

***Response:*** *We have corrected this word in Fig. 10 c, d*

P36, Figure 15 a,b: Frequency is misspelled

***Response:*** *We have corrected this word in Fig. 10 c, d*

[revised manuscript text omitted]